# PERFORMANCE GAPS IN MULTI-VIEW CLUSTERING UNDER THE NESTED MATRIX-TENSOR MODEL

**Hugo Lebeau**[1*]    **Mohamed El Amine Seddik**[2]    **José Henrique de Morais Goulart**[3]

[1]Univ. Grenoble Alpes, CNRS, Inria, Grenoble INP, LIG    [2]Technology Innovation Institute
[3]Univ. Toulouse, INP-ENSEEIHT, IRIT, CNRS
*hugo.lebeau@univ-grenoble-alpes.fr

## ABSTRACT

We study the estimation of a planted signal hidden in a recently introduced nested matrix-tensor model, which is an extension of the classical spiked rank-one tensor model, motivated by multi-view clustering. Prior work has theoretically examined the performance of a tensor-based approach, which relies on finding a best rank-one approximation, a problem known to be computationally hard. A tractable alternative approach consists in computing instead the best rank-one (matrix) approximation of an unfolding of the observed tensor data, but its performance was hitherto unknown. We quantify here the performance gap between these two approaches, in particular by deriving the precise algorithmic threshold of the unfolding approach and demonstrating that it exhibits a BBP-type transition behavior (Baik et al., 2005). This work is therefore in line with recent contributions which deepen our understanding of why tensor-based methods surpass matrix-based methods in handling structured tensor data.

## 1 INTRODUCTION

In the age of artificial intelligence, handling vast amounts of data has become a fundamental aspect of machine learning tasks. Datasets are often high-dimensional and composed of multiple modes, such as various modalities, sensors, sources, types, or domains, naturally lending themselves to be represented as tensors. Tensors offer a richer structure compared to traditional one-dimensional vectors and two-dimensional matrices, making them increasingly relevant in various applications, including statistical learning and data analysis (Landsberg, 2012; Sun et al., 2014).

Yet, in the existing literature, there is a notable scarcity of theoretical studies that specifically address the performance gaps between tensor-based methods and traditional (matrix) spectral methods in the context of high-dimensional data analysis. While tensor methods have shown promise in various applications, including multi-view clustering, co-clustering, community detection, and latent variable modeling (Wu et al., 2019; Anandkumar et al., 2014; Papalexakis et al., 2012; Wang et al., 2023), little attention has been devoted to rigorously quantifying the advantages and drawbacks of leveraging the hidden low-rank tensor structure. Filling this gap by conducting a thorough theoretical analysis is crucial for gaining a deeper understanding of the practical implications and potential performance gains associated with tensor-based techniques.

In the specific case of multi-view clustering, Seddik et al. (2023a) recently proposed a spectral tensor method and carried out a precise analysis of its performance in the large-dimensional limit. Their method consists in computing a best rank-one (tensor) approximation of a *nested matrix-tensor model*, which, in particular, generalizes the classical *rank-one spiked tensor model* of Montanari & Richard (2014), and can be described as follows. Assume that we observe $m$ transformations of a $p \times n$ matrix $\mathbf{M} = \boldsymbol{\mu}\boldsymbol{y}^\top + \mathbf{Z}$ representing $n$ points in dimension $p$ split into two clusters centered around $\pm\boldsymbol{\mu}$, with $\boldsymbol{y} \in \{-1, 1\}^n$ and $\mathbf{Z}$ a Gaussian matrix encoding the "inherent" dispersion of individuals (that is, regardless of measurement errors) around the center of their respective cluster. Mathematically, each view is thus expressed as

$$\mathbf{X}_k = f_k(\boldsymbol{\mu}\boldsymbol{y}^\top + \mathbf{Z}) + \mathbf{W}_k, \quad k = 1, \ldots, m, \tag{1}$$

where $f_k$ models the transformation applied to $\mathbf{M}$ on the $k$-th view and $\mathbf{W}_k$ is an additive observation noise with i.i.d. entries drawn from $\mathcal{N}(0, 1)$. The nested-matrix tensor model then arises when

we take $f_k(\mathbf{M}) = h_k \mathbf{M}$, meaning the function $f_k$ simply rescales the matrix $\mathbf{M}$ by an unknown coefficient $h_k \in \mathbb{R}$. With $\boldsymbol{h} = (h_1 \quad \dots \quad h_m)^\top \in \mathbb{R}^m$ and $\otimes$ denoting the outer product, this gives

$$\mathbf{X} = (\boldsymbol{\mu}\boldsymbol{y}^\top + \mathbf{Z}) \otimes \boldsymbol{h} + \mathbf{W} \in \mathbb{R}^{p \times n \times m}. \qquad \text{(Nested Matrix-Tensor Model)}$$

By seeking a best rank-one approximation of $\mathbf{X}$ to estimate its latent clustering structure, Seddik et al. (2023a) showed *empirically* that it outperforms an unfolding approach based on applying an SVD to an *unfolding* of $\mathbf{X}$ (Ben Arous et al., 2021; Lebeau et al., 2024), which is a matrix obtained by rearranging the entries of a tensor (see Section 2.2). However, the tensor-based approach hinges upon solving a problem which is worst-case NP-hard, unlike the unfolding approach. A natural question is thus: what is the exact performance gap that exists between these two approaches, as a function of some measure of difficulty of the problem (typically, a measure of signal-to-noise ratio)?

Here, in order to answer this question, we rigorously study the unfolding method by deploying tools from random matrix theory. Specifically, our main contributions are

- within the framework of the general nested matrix-tensor model, we derive the limiting spectral distribution of the unfoldings of the tensor (Theorems 1 and 3) and precisely quantify how well the hidden low-rank (tensor) structure can be recovered from them in the high-dimensional regime (Theorems 2 and 7);
- we perform a similar random matrix analysis of the model when the vector spanning the third mode is known (Theorems 3 and 4), providing an optimal upper bound on the recovery performance;
- in the context of multi-view clustering, we compare the performance of the tensor and unfolding approaches to the optimal one and specify the gap between them thanks to our theoretical findings (Theorem 5), supported by empirical results[1].

Although the above described model arises from a rather particular choice of view transformations $f_k$, it is amenable to a precise estimation performance analysis, either by means of a tensor spectral estimator as recently done by Seddik et al. (2023a), or via a matrix spectral estimator as we consider in the present paper. Moreover, from a broader perspective (that is, beyond the multi-view clustering problem considered here), this model can be viewed as a more flexible version of the rank-one spiked model, incorporating a nested structure that allows for versatile data modeling, deviating from a pure rank-one assumption. A common low-rank structure encoding the underlying latent clustering pattern is shared by all slices $\mathbf{X}_k$, which represent distinct views of the data. In particular, when the variances of the elements in $\mathbf{Z}$ approach zero, the rank-one spiked model is retrieved. Hence, we believe that the nested tensor-matrix model (and extensions) can be a useful tool in other contexts in the broader area of statistical learning.

**Related work.** In the machine learning literature, the notion of "view" is fairly general and models data whose form may differ but all represent the same object seen from different (and complementary) angles (for instance, multiple descriptors of an image, translations of a text or features of a webpage such as its hyperlinks, text and images). Various approaches have been considered to address multi-view clustering problems. For instance, Nie et al. (2016; 2017b;a) consider a graph-based model and construct a similarity matrix by integrating all views with a weighted sum before applying spectral clustering. Other approaches, relying upon a space-learning-based model (Wang et al., 2017; Zhang et al., 2017; Wang et al., 2019; Peng et al., 2019), reconstruct the data in an ideal space where clustering is easy. Zhang et al. (2019) suggest a method which is more suitable for large datasets by mixing binary coding and clustering. Tensor methods have also been considered: Wu et al. (2019) propose an essential tensor learning approach for Markov chain-based multi-view spectral clustering. Furthermore, Liu et al. (2021; 2023) design simple yet effective methods for multi-view clustering relying on multiple kernel $k$-means. Our work differs from these previous contributions in that it is focused on a tensor model having the specific form of equation Nested Matrix-Tensor Model. Even though this model corresponds to a rather particular case of the general setting given in equation 1, our results represent a first step towards precisely understanding how tensor methods can contribute to addressing the latter.

---

[1]Note that these numerical results are only meant to illustrate our theoretical findings, showing their implications in practice. However, our work does not purport to explain the performance gap between *any* tensor-based and *any* matrix-based multi-view clustering methods.

Regarding the analysis of performance gaps between tensor- and matrix-based methods, we can mention the recent work by Seddik et al. (2023b), where the authors proposed a data model that consists of a Gaussian mixture assuming a low-rank tensor structure on the population means and further characterized the theoretical performance gap between a simple tensor-based method and a flattening-based method that neglects the low-rank structure. Their study has demonstrated that the tensor approach yields provably better performance compared to treating the data as mere vectors.

**Proofs and simulations.** All proofs are deferred to the appendix. Python codes to reproduce simulations are available in the following GitHub repository `https://github.com/HugoLeb eau/nested_matrix-tensor`.

## 2 TENSORS AND RANDOM MATRIX THEORY

### 2.1 GENERAL NOTATIONS

The symbols $a$, $\boldsymbol{a}$ and $\boldsymbol{A}$ respectively denote a scalar, a vector and a matrix. Their random counterparts are $\mathsf{a}$, $\mathbf{a}$ and $\mathbf{A}$, respectively. Tensors (be they random or not) are denoted $\mathsf{A}$. The set of integers $\{1, \ldots, n\}$ is denoted $[n]$. The unit sphere in $\mathbb{R}^n$ is $\mathbb{S}^{n-1} = \{\boldsymbol{x} \in \mathbb{R}^n \mid \|\boldsymbol{x}\| = 1\}$. The set of eigenvalues of a matrix $\boldsymbol{A}$ is called its spectrum and denoted $\mathrm{sp}(\boldsymbol{A})$. The support of a measure $\mu$ is denoted $\mathrm{supp}\,\mu$. As usual, $\delta_x$ is the Dirac measure at point $x$. Given two sequences of scalars $f_n$ and $g_n$, the notation $f_n = \Theta(g_n)$ means that there exist constants $C, C' > 0$ and $n_0$ such that $n \geqslant n_0 \implies C\,|g_n| \leqslant |f_n| \leqslant C'\,|g_n|$. The convergence in distribution of a sequence of random variables $(\mathsf{x}_n)_{n \geqslant 0}$ is denoted $\mathsf{x}_n \xrightarrow[n \to +\infty]{\mathcal{D}} \mathcal{L}$ where $\mathcal{L}$ is the limiting distribution.

### 2.2 TENSORS AND RELATED OPERATIONS

For our purposes, we can think of tensors as multidimensional arrays. In this work, we will only consider tensors of order 3, i.e., elements of $\mathbb{R}^{n_1 \times n_2 \times n_3}$. For such a tensor $\mathsf{T}$ and $(i, j, k) \in [n_1] \times [n_2] \times [n_3]$, $\mathsf{T}_{i,j,k}$ denotes its $(i, j, k)$-entry. $\mathsf{T}$ can be *unfolded* along one of its three modes to construct a "matricized version" of the tensor: the unfolding of $\mathsf{T}$ along mode 1 is the matrix $\boldsymbol{T}^{(1)} \in \mathbb{R}^{n_1 \times n_2 n_3}$ such that $\mathsf{T}_{i,j,k} = T^{(1)}_{i,n_3(j-1)+k}$, and likewise for $\boldsymbol{T}^{(2)}$ and $\boldsymbol{T}^{(3)}$ — the unfoldings of $\mathsf{T}$ along modes 2 and 3. The inner product between two tensors $\mathsf{T}, \mathsf{T}' \in \mathbb{R}^{n_1 \times n_2 \times n_3}$ is $\langle \mathsf{T}, \mathsf{T}' \rangle = \sum_{i,j,k=1}^{n_1,n_2,n_3} \mathsf{T}_{i,j,k} \mathsf{T}'_{i,j,k}$ and the Frobenius norm of $\mathsf{T}$ is defined simply as $\|\mathsf{T}\|_{\mathrm{F}} = \sqrt{\langle \mathsf{T}, \mathsf{T} \rangle}$.

The *outer product* $\otimes$ allows to construct an order-3 tensor from a matrix $\boldsymbol{A} \in \mathbb{R}^{n_1 \times n_2}$ and a vector $\boldsymbol{w} \in \mathbb{R}^{n_3}$, $[\boldsymbol{A} \otimes \boldsymbol{w}]_{i,j,k} = A_{i,j} w_k$, or from three vectors $(\boldsymbol{u}, \boldsymbol{v}, \boldsymbol{w}) \in \mathbb{R}^{n_1} \times \mathbb{R}^{n_2} \times \mathbb{R}^{n_3}$, $[\boldsymbol{u} \otimes \boldsymbol{v} \otimes \boldsymbol{w}]_{i,j,k} = u_i v_j w_k$. In the latter case, the tensor is said to be rank-one.

Unfoldings of tensors defined with outer products are often expressed using *Kronecker products* $\boxtimes$. Given two matrices (or vectors if the second dimension is set to 1) $\boldsymbol{A} \in \mathbb{R}^{n_1 \times n_2}, \boldsymbol{B} \in \mathbb{R}^{p_1 \times p_2}$, their Kronecker product $\boldsymbol{A} \boxtimes \boldsymbol{B}$ is the $n_1 p_1 \times n_2 p_2$ matrix such that $[\boldsymbol{A} \boxtimes \boldsymbol{B}]_{p_1(i-1)+r, p_2(j-1)+s} = A_{i,j} B_{r,s}$. Among the numerous properties of the Kronecker product, we highlight the fact that it is bilinear, associative, $(\boldsymbol{A} \boxtimes \boldsymbol{B})^\top = \boldsymbol{A}^\top \boxtimes \boldsymbol{B}^\top$ and $(\boldsymbol{A} \boxtimes \boldsymbol{B})(\boldsymbol{C} \boxtimes \boldsymbol{D}) = \boldsymbol{AC} \boxtimes \boldsymbol{BD}$.

*Example.* With $\boldsymbol{a} = \begin{bmatrix} \boldsymbol{A}_{1,:} & \cdots & \boldsymbol{A}_{n_1,:} \end{bmatrix}^\top \in \mathbb{R}^{n_1 n_2}$,

$$
\begin{aligned}
[\boldsymbol{A} \otimes \boldsymbol{w}]^{(1)} &= \boldsymbol{A}\left(\boldsymbol{I}_{n_2} \boxtimes \boldsymbol{w}\right)^\top, & [\boldsymbol{u} \otimes \boldsymbol{v} \otimes \boldsymbol{w}]^{(1)} &= \boldsymbol{u}\left(\boldsymbol{v} \boxtimes \boldsymbol{w}\right)^\top, \\
[\boldsymbol{A} \otimes \boldsymbol{w}]^{(2)} &= \boldsymbol{A}^\top\left(\boldsymbol{I}_{n_1} \boxtimes \boldsymbol{w}\right)^\top, & [\boldsymbol{u} \otimes \boldsymbol{v} \otimes \boldsymbol{w}]^{(2)} &= \boldsymbol{v}\left(\boldsymbol{u} \boxtimes \boldsymbol{w}\right)^\top, \\
[\boldsymbol{A} \otimes \boldsymbol{w}]^{(3)} &= \boldsymbol{w}\boldsymbol{a}^\top, & [\boldsymbol{u} \otimes \boldsymbol{v} \otimes \boldsymbol{w}]^{(3)} &= \boldsymbol{w}\left(\boldsymbol{u} \boxtimes \boldsymbol{v}\right)^\top.
\end{aligned}
$$

### 2.3 RANDOM MATRIX TOOLS

The results presented below are derived using tools from the theory of large random matrices (Bai & Silverstein, 2010; Pastur & Shcherbina, 2011; Couillet & Liao, 2022), the main tools of which are recalled here.

Given a random symmetric matrix $\mathbf{S} \in \mathbb{R}^{n \times n}$, we are interested in the behavior of its (real) eigenvalues and eigenvectors as $n \to +\infty$[2]. A first kind of result is the weak convergence of its *empirical spectral distribution* (ESD) $\frac{1}{n} \sum_{\lambda \in \mathrm{sp}(\mathbf{S})} \delta_\lambda$ towards a *limiting spectral distribution* (LSD) $\mu$. The latter is often characterized by its Stieltjes transform $m_\mu : s \in \mathbb{C} \setminus \mathrm{sp}\,\mathbf{S} \mapsto \int_{\mathbb{R}} \frac{\mu(\mathrm{d}t)}{t-s}$, from which $\mu$ can be recovered using the inverse formula $\mu(\mathrm{d}t) = \lim_{\eta \to 0} \frac{1}{\pi} \Im[m_\mu(t + i\eta)]$. A second kind of result is the almost sure convergence of the alignment between a vector $\boldsymbol{x} \in \mathbb{R}^n$ and an eigenvector $\hat{\mathbf{x}}$ of $\mathbf{S}$, i.e., the quantity $\left|\boldsymbol{x}^\top \hat{\mathbf{x}}\right|^2$, towards a fixed value in $[0, 1]$.

A central tool to derive such results is the resolvent $\mathbf{Q_S}(s) = (\mathbf{S} - s\boldsymbol{I}_n)^{-1}$, defined for all $s \in \mathbb{C} \setminus \mathrm{sp}\,\mathbf{S}$. Indeed, $\frac{1}{n} \mathrm{Tr}\,\mathbf{Q_S}(s)$ is the Stieltjes transform evaluated at $s$ of the ESD of $\mathbf{S}$, so studying its asymptotic behavior shall give us insight into $m_\mu$. The alignments can as well be studied through the resolvent thanks to the property $\left|\boldsymbol{x}^\top \hat{\mathbf{x}}\right|^2 = -\frac{1}{2i\pi} \oint_\gamma \boldsymbol{x}^\top \mathbf{Q_S}(s)\boldsymbol{x}\,\mathrm{d}s$ where $\gamma$ is a positively-oriented complex contour circling around the eigenvalue associated to $\hat{\mathbf{x}}$ (assuming it has multiplicity one) and leaving all other eigenvalues outside.

Because of the random nature of $\mathbf{Q_S}$, we will first seek a deterministic equivalent, i.e., a *deterministic* matrix $\bar{\mathbf{Q}}_\mathbf{S}$ such that both quantities $\frac{1}{n} \mathrm{Tr}\,\boldsymbol{A}\left(\mathbf{Q_S} - \bar{\mathbf{Q}}_\mathbf{S}\right)$ and $\boldsymbol{a}^\top \left(\mathbf{Q_S} - \bar{\mathbf{Q}}_\mathbf{S}\right) \boldsymbol{b}$ vanish almost surely as $n \to +\infty$ for any (sequences of) deterministic matrices $\boldsymbol{A}$ and vectors $\boldsymbol{a}, \boldsymbol{b}$ of bounded norms (respectively, operator and Euclidean). This is denoted $\mathbf{Q_S} \longleftrightarrow \bar{\mathbf{Q}}_\mathbf{S}$. The following lemma will be extensively used to derive such equivalents.

**Lemma 1** (Stein (1981)). *Let* $\mathrm{z} \sim \mathcal{N}(0, 1)$ *and* $f : \mathbb{R} \to \mathbb{R}$ *be a continuously differentiable function. When the following expectations exist,* $\mathbb{E}[\mathrm{z}f(\mathrm{z})] = \mathbb{E}[f'(\mathrm{z})]$.

## 3  MAIN RESULTS

Before presenting its practical applications in Section 4, we recall the definition of the nested matrix-tensor model in a general framework. Consider the following statistical model.

$$\mathbf{T} = \beta_T \mathbf{M} \otimes \boldsymbol{z} + \frac{1}{\sqrt{n_T}}\mathbf{W} \in \mathbb{R}^{n_1 \times n_2 \times n_3}, \qquad \mathbf{M} = \beta_M \boldsymbol{x}\boldsymbol{y}^\top + \frac{1}{\sqrt{n_M}}\mathbf{Z} \in \mathbb{R}^{n_1 \times n_2}, \qquad (2)$$

where $n_M = n_1 + n_2$ and $n_T = n_1 + n_2 + n_3$, $\boldsymbol{x}, \boldsymbol{y}$ and $\boldsymbol{z}$ are of unit norm and the entries of $\mathbf{W}$ and $\mathbf{Z}$ are independent Gaussian random variables[3]: $\mathsf{W}_{i,j,k} \overset{\text{i.i.d.}}{\sim} \mathcal{N}(0, 1)$, $\mathsf{Z}_{i,j} \overset{\text{i.i.d.}}{\sim} \mathcal{N}(0, 1)$. $\mathbf{M}$ is a rank-1 signal $\beta_M \boldsymbol{x}\boldsymbol{y}^\top$ corrupted by noise $\mathbf{Z}$, modelling the data matrix, whereas $\mathbf{T}$ models its multi-view observation $\beta_T \mathbf{M} \otimes \boldsymbol{z}$ corrupted by noise $\mathbf{W}$. The positive parameters $\beta_M$ and $\beta_T$ control the signal-to-noise ratio (SNR). Our interest is the statistical recovery of $\boldsymbol{x}, \boldsymbol{y}$ or $\boldsymbol{z}$ in the regime where $n_1, n_2, n_3 \to +\infty$ with $0 < c_\ell = \lim n_\ell / n_T < 1$ for all $\ell \in [3]$. This models the fact that, in practice, we deal with large tensors whose dimensions have comparable sizes.

Seddik et al. (2023a) have studied the spectral estimator of $\boldsymbol{x}, \boldsymbol{y}$ and $\boldsymbol{z}$ based on computing the best rank-one approximation of $\mathbf{T}$, that is, by solving

$$(\hat{\mathbf{x}}, \hat{\mathbf{y}}, \hat{\mathbf{z}}) = \underset{(\boldsymbol{u}, \boldsymbol{v}, \boldsymbol{w}) \in \mathbb{S}^{n_1 - 1} \times \mathbb{S}^{n_2 - 1} \times \mathbb{S}^{n_3 - 1}}{\arg\max} \langle \mathbf{T}, \boldsymbol{u} \otimes \boldsymbol{v} \otimes \boldsymbol{w} \rangle.$$

Concretely, they used random matrix tools to assess its performance in the recovery of $\boldsymbol{x}, \boldsymbol{y}$ and $\boldsymbol{z}$, by deploying a recent approach developed by Goulart et al. (2022); Seddik et al. (2022). In this work, we study instead the performance of a spectral approach based on computing the dominant singular vectors of the (matrix) *unfoldings* of $\mathbf{T}$, aiming to precisely quantify the performance gap between these different approaches.

Because $\mathbf{M}$ has the structure of a standard spiked matrix model (Benaych-Georges & Nadakuditi, 2011; Couillet & Liao, 2022) with a rank-one perturbation $\beta_M \boldsymbol{x}\boldsymbol{y}^\top$ of a random matrix $\frac{1}{\sqrt{n_m}}\mathbf{Z}$, we shall assume $\beta_M = \Theta(1)$ since we know that it is in this "non-trivial regime" that the recovery of $\boldsymbol{x}$

---

[2]Strictly speaking, we consider a *sequence* of matrices $\mathbf{S}_n$ but the dependence on $n$ is dropped as the assumption $n \to +\infty$ models the fact that, in practice, $n$ is large.

[3]The "interpolation trick" of Lytova & Pastur (2009, Corollary 3.1) allows to extend these results to non-Gaussian noise up to a control on the moments of the distribution.

or $y$ given $M$ is neither too easy (too high SNR) nor too hard (too small SNR) and a phase-transition phenomenon (Baik et al., 2005) between impossible and possible recovery can be observed. However, we shall see that the algorithmic phase transition related to the unfolding approach takes place when $\beta_T = \Theta(n_T^{1/4})$ in *lieu* of $\beta_T = \Theta(1)$. This is different from the tensor spectral approach for which both $\beta_M$ and $\beta_T$ are $\Theta(1)$ in the non-trivial regime, supposing a better performance of the latter method, although, practically, no known algorithm is able to compute it below $\beta_T = \Theta(n_T^{1/4})$ (Montanari & Richard, 2014).

## 3.1 Unfoldings Along the First Two Modes

We start by studying the recovery of $y$ (resp. $x$) from the unfolding $\mathbf{T}^{(2)}$ (resp. $\mathbf{T}^{(1)}$). In a multi-view clustering perspective — which motivates our work and which will be developed in Section 4 — we are especially interested in the recovery of $y$ since it carries the class labels. Therefore, we present our results for $\mathbf{T}^{(2)}$ only. As $x$ and $y$ play a symmetric role, it is easy to deduce the results for $\mathbf{T}^{(1)}$ from those presented below. The recovery of $z$ from $\mathbf{T}^{(3)}$ is dealt with in Appendix A.

Following the model presented in equation 2, the unfolding along the second mode of $\mathbf{T}$ develops as

$$\mathbf{T}^{(2)} = \beta_T \beta_M \boldsymbol{y} (\boldsymbol{x} \boxtimes \boldsymbol{z})^\top + \frac{\beta_T}{\sqrt{n_M}} \mathbf{Z}^\top (\boldsymbol{I}_{n_1} \boxtimes \boldsymbol{z})^\top + \frac{1}{\sqrt{n_T}} \mathbf{W}^{(2)}. \tag{3}$$

Hence, a natural estimator $\hat{\mathbf{y}}$ of $y$ is the dominant left singular vector of $\mathbf{T}^{(2)}$ or, equivalently, the dominant eigenvector of $\mathbf{T}^{(2)}\mathbf{T}^{(2)\top}$. The latter being symmetric, it is better suited to the tools presented in Section 2.3. Our first step is to characterize the limiting spectral distribution of this random matrix. However, one must be careful with the fact that the dimensions of $\mathbf{T}^{(2)} \in \mathbb{R}^{n_2 \times n_1 n_3}$ do not have sizes of the same order, causing the spectrum of $\mathbf{T}^{(2)}\mathbf{T}^{(2)\top}$ to diverge as $n_1, n_2, n_3 \to +\infty$. In fact, its eigenvalues gather in a "bulk" centered around a $\Theta(n_T)$ value and spread on an interval of size $\Theta(\sqrt{n_T})$ — a phenomenon which was first characterized by Ben Arous et al. (2021). For this reason, in Theorem 1, we do not specify the LSD of $\mathbf{T}^{(2)}\mathbf{T}^{(2)\top}$ *per se* but of a properly centered-and-scaled version of it, whose spectrum no longer diverges. Moreover, it is expected that the rank-one signal $\beta_T \beta_M \boldsymbol{y} (\boldsymbol{x} \boxtimes \boldsymbol{z})^\top$ causes the presence of an isolated eigenvalue in the spectrum of $\mathbf{T}^{(2)}\mathbf{T}^{(2)\top}$ with corresponding eigenvector positively correlated with $y$ when it is detectable, i.e., when $\beta_T \beta_M$ is large enough.

Our second step is thus to precisely specify what is meant by "large enough" and characterize the asymptotic position of this spike eigenvalue and the alignment with $y$ of its corresponding eigenvector. It turns out that the signal vanishes if $\beta_T$ does not scale with $n_T$. Precisely, $\beta_T^2 n_T / \sqrt{n_1 n_2 n_3}$ must converge to a fixed positive quantity[4], denoted $\rho_T$, to reach the "non-trivial regime" — that is, one where the signal and the noise in the model have comparable strengths. However, because the noise in $\mathbf{Z}$ is also weighted by $\beta_T$, this affects the shape of the bulk. Hence, the value of $\rho_T$ influences the limiting spectral distribution of $\mathbf{T}^{(2)}\mathbf{T}^{(2)\top}$ and shall appear in its defining equation[5]. Having said all this, we are now ready to introduce the following theorem.

**Theorem 1** (Limiting Spectral Distribution)**.** *As $n_1, n_2, n_3 \to +\infty$, the centered-and-scaled matrix $\frac{n_T}{\sqrt{n_1 n_2 n_3}} \mathbf{T}^{(2)}\mathbf{T}^{(2)\top} - \frac{n_2 + n_1 n_3}{\sqrt{n_1 n_2 n_3}} \boldsymbol{I}_{n_2}$ has a limiting spectral distribution $\tilde{\nu}$ whose Stieltjes transform $\tilde{m}(\tilde{s})$ is solution to*

$$\frac{\rho_T c_2}{1 - c_3} \tilde{m}^3(\tilde{s}) + \left(1 + \tilde{s} \frac{\rho_T c_2}{1 - c_3}\right) \tilde{m}^2(\tilde{s}) + \left(\tilde{s} + \frac{\rho_T (c_2 - c_1)}{1 - c_3}\right) \tilde{m}(\tilde{s}) + 1 = 0, \qquad \tilde{s} \in \mathbb{C} \setminus \operatorname{supp} \tilde{\nu}$$

*where $\rho_T = \lim \frac{\beta_T^2 n_T}{\sqrt{n_1 n_2 n_3}}$.*

*Proof.* See Appendix B. □

---

[4]In fact, $\beta_T^2 n_T / \sqrt{n_1 n_2 n_3}$ only needs to stay bounded between two *strictly positive* quantities, i.e., stay $\Theta(1)$. It is however easier for the present analysis to consider that it converges (as it allows to express the LSD) and this assumption has no practical implications.

[5]This may be at first surprising because $\rho_T$ relates to the strength of the *signal* while the LSD stems from the *noise*. We see that $\mathbf{Z}$ plays an ambivalent role of both a signal and a noise term.

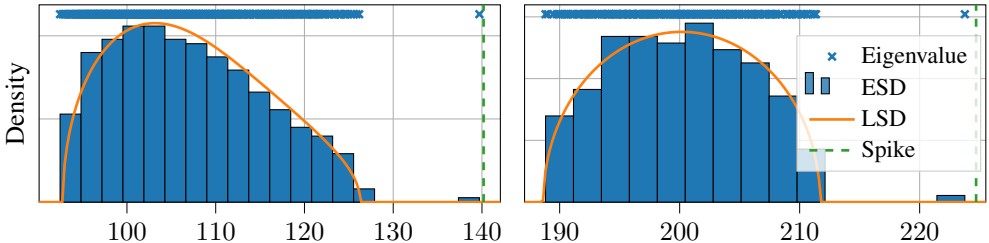

Figure 1: **Empirical Spectral Distribution (ESD)** and **Limiting Spectral Distribution (LSD)** of $\mathbf{T}^{(2)}\mathbf{T}^{(2)\top}$ (**left**) and $\mathbf{T}^{(3)}\mathbf{T}^{(3)\top}$ (**right**) with $n_1 = 600$, $n_2 = 400$ and $n_3 = 200$. Both spectra show an **isolated eigenvalue** close to its predicted asymptotic position, represented by the green dashed line. **Left**: $\rho_T = 2$, $\beta_M = 1.5$. The centered-and-scaled LSD $\tilde{\nu}$ and spike location $\tilde{\xi}$ are defined in Theorems 1 and 2. **Right**: $\varrho = 4$, $\beta_M = 3$. The LSD is a shifted-and-rescaled semi-circle distribution and the normalized spike location is $\varrho + \frac{1}{\varrho}$ as precised in Theorems 6 and 7.

As mentioned in Section 2.3, the LSD $\tilde{\nu}$ is characterized by its Stieltjes transform, uniquely defined as the solution of a polynomial equation[6]. The influence of $\rho_T$ on the LSD of $\mathbf{T}^{(2)}\mathbf{T}^{(2)\top}$ is made explicit in this equation and it is interesting to remark that, if $\rho_T = 0$ (i.e., in the absence of signal), this equality reduces to $\tilde{m}^2(\tilde{s}) + \tilde{s}\tilde{m}(\tilde{s}) + 1 = 0$, which is a well-known characterization of the Stieltjes transform of the semi-circle distribution (Pastur & Shcherbina, 2011, Corollary 2.2.8). Note also that the condition $\rho_T = \Theta(1)$ amounts to saying that $\beta_T = \Theta(n_T^{1/4})$, which coincides with the conjectured "computational threshold" under which no known algorithm is able to detect a signal without prior information (Montanari & Richard, 2014; Ben Arous et al., 2021).

The ESD and LSD of $\mathbf{T}^{(2)}\mathbf{T}^{(2)\top}$ with parameters $(\rho_T, \beta_M) = (2, 1.5)$ are represented in the left panel of Figure 1. We observe a good agreement between the actual and predicted shape of the bulk. As expected, we see an isolated eigenvalue on the right which only appears for sufficiently high values of $\rho_T$ and $\beta_M$. The following theorem specifies this behavior and quantifies the alignment between the signal $\mathbf{y}$ and the spike eigenvector $\hat{\mathbf{y}}$.

**Theorem 2** (Spike Behavior)**.**

*Let*
$$\tilde{\xi} = \frac{\rho_T}{\beta_M^2}\left(\frac{c_1}{1-c_3} + \beta_M^2\right)\left(\frac{c_2}{1-c_3} + \beta_M^2\right) + \frac{1}{\rho_T\left(\frac{c_2}{1-c_3} + \beta_M^2\right)}$$

*and*
$$\zeta = 1 - \frac{1}{\beta_M^2\left(\frac{c_2}{1-c_3} + \beta_M^2\right)}\left[\left(\frac{\beta_M^2}{\rho_T\left(\frac{c_2}{1-c_3} + \beta_M^2\right)}\right)^2 + \frac{c_2}{1-c_3}\left(\frac{c_1}{1-c_3} + \beta_M^2\right)\right].$$

*If $\zeta > 0$, then the centered-and-scaled matrix $\frac{n_T}{\sqrt{n_1 n_2 n_3}}\mathbf{T}^{(2)}\mathbf{T}^{(2)\top} - \frac{n_2+n_1 n_3}{\sqrt{n_1 n_2 n_3}}\mathbf{I}_{n_2}$ has an isolated eigenvalue asymptotically located in $\tilde{\xi}$ almost surely. Furthermore, in this case, the alignment between the corresponding eigenvector $\hat{\mathbf{y}}$ and the true signal $\mathbf{y}$ converges to $\zeta$ almost surely, i.e.,*

$$\left|\hat{\mathbf{y}}^\top \mathbf{y}\right|^2 \xrightarrow[n_1,n_2,n_3\to+\infty]{} \zeta \qquad \text{almost surely.}$$

*Proof.* See Appendix C. □

Naturally, we must assume $\rho_T, \beta_M > 0$ for $\tilde{\xi}$ and $\zeta$ to be well defined. The location of the isolated eigenvalue in the spectrum of $\mathbf{T}^{(2)}\mathbf{T}^{(2)\top}$ predicted from the expression of $\tilde{\xi}$ is represented as the green dashed line in the left panel of Figure 1. In fact, Theorem 2 reveals a phase transition phenomenon between impossible and possible recovery of the signal with the estimator $\hat{\mathbf{y}}$. This is

---

[6]Although this is not the only solution to this equation, it is the only one that has the properties of a Stieltjes transform, such as $\Im[\tilde{s}]\Im[\tilde{m}(\tilde{s})] > 0$ for all $\tilde{s} \in \mathbb{C} \setminus \mathbb{R}$ (see, e.g., Tao (2012) for other properties).

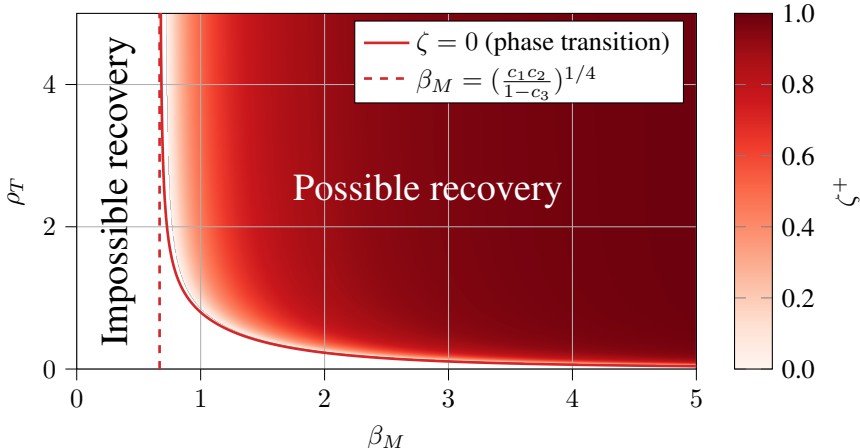

Figure 2: **Asymptotic alignment** $\zeta^+ = \max(\zeta, 0)$ between the signal $\boldsymbol{y}$ and the dominant eigenvector of $\mathbf{T}^{(2)}\mathbf{T}^{(2)\top}$, as defined in Theorem 2, with $c_1 = \frac{1}{2}$, $c_2 = \frac{1}{3}$ and $c_3 = \frac{1}{6}$. The curve $\zeta = 0$ is the position of the **phase transition** between the impossible detectability of the signal (below) and the presence of an isolated eigenvalue in the spectrum of $\mathbf{T}^{(2)}\mathbf{T}^{(2)\top}$ with corresponding eigenvector correlated with the signal (above). It has an asymptote $\beta_M = (\frac{c_1 c_2}{1-c_3})^{1/4}$, represented by the red dashed line, as $\rho_T \to +\infty$.

precisely quantified by the value of $\zeta^+ = \max(\zeta, 0)$: the closer it is to 1, the better is the estimation of $\boldsymbol{y}$. The precise dependence of $\zeta$ on $\rho_T$ and $\beta_M$ is hard to interpret directly from its expression. Figure 2 displays $\zeta^+$ as a function of $\rho_T$ and $\beta_M$. The expression of the curve $\zeta = 0$ marking the position of the transition from impossible to possible recovery is given by the following proposition.

**Proposition 1** (Phase Transition). *If* $\beta_M^4 > \frac{c_1 c_2}{1-c_3}$, *then* $\zeta = 0 \iff \rho_T = \dfrac{\beta_M^2}{\left(\frac{c_2}{1-c_3} + \beta_M^2\right)\sqrt{\beta_M^4 - \frac{c_1 c_2}{1-c_3}}}$.

We see that, if $\beta_M^4 \leqslant \frac{c_1 c_2}{1-c_3}$, it is impossible to find $\rho_T > 0$ such that $\zeta > 0$. This is due to the fact that $\beta_M^4 = \frac{c_1 c_2}{1-c_3}$ corresponds to the position of the phase transition in the estimation of $\boldsymbol{y}$ *from* $\mathbf{M}$. If the signal is not detectable from $\mathbf{M}$, there is obviously no chance to recover it from $\mathbf{T}$. Moreover, as $\rho_T$ grows, the value of $\beta_M$ such that $\zeta = 0$ coincides with $(\frac{c_1 c_2}{1-c_3})^{1/4}$ but it goes to $+\infty$ as $\rho_T$ approaches 0. This shows the importance of having $\beta_T = \Theta(n_T^{1/4})$ (in which case it is more convenient to work with the rescaled version $\rho_T$ of $\beta_T$): if $\beta_T$ is an order below ($\rho_T \to 0$) then we are stuck in the "Impossible recovery" zone while if $\beta_T$ is an order above ($\rho_T \to +\infty$) then estimating from $\mathbf{T}$ is just like estimating from $\mathbf{M}$. It is precisely in the regime $\beta_T = \Theta(n_T^{1/4})$ that this phase-transition phenomenon can be observed, thereby justifying its designation as "non-trivial".

*Remark.* It should be noted that the aforementioned impossibility of (partially) recovering the sought signal in a given regime refers only to the case where such a recovery is carried out by the unfolding method. In other words, our discussion concerns algorithmic thresholds pertaining to such method, and not statistical ones.

## 3.2 ESTIMATION WITH WEIGHTED MEAN

Before diving into the application of the previous results to multi-view clustering (where we will be interested in the estimation of the class labels contained in $\boldsymbol{y}$), we propose an analysis of a related matrix model corresponding to the optimal estimation of $\boldsymbol{y}$ when $\boldsymbol{z}$ is perfectly known. These results will give us an optimistic upper bound on the performance of the estimation of $\boldsymbol{y}$ from $\mathbf{T}$.

In case $\boldsymbol{z}$ is known, $\boldsymbol{y}$ can be estimated with the following weighted mean of $\mathbf{T}$ along mode 3,

$$\bar{\mathbf{T}} = \sum_{k=1}^{n_3} z_k \mathbf{T}_{:,:,k} = \beta_T \beta_M \boldsymbol{x}\boldsymbol{y}^\top + \frac{\varsigma}{\sqrt{n_M}}\mathbf{N} \tag{4}$$

where $N_{i,j} \overset{\text{i.i.d.}}{\sim} \mathcal{N}(0,1)$ and $\varsigma^2 = \beta_T^2 + \frac{n_M}{n_T}$. It is well known that the dominant right singular vector of $\bar{\mathbf{T}}$ is an optimal estimator of $\boldsymbol{y}$ under this model (Onatski et al., 2013; Löffler et al., 2020). Hence, we study the spectrum of $\frac{1}{\varsigma^2} \bar{\mathbf{T}}^\top \bar{\mathbf{T}}$, which is a sample covariance matrix — a standard model in random matrix theory (Pastur & Shcherbina, 2011; Bai & Silverstein, 2010). Its eigenvalue distribution converges to the Marchenko-Pastur distribution (Marčenko & Pastur, 1967), as expressed in the following theorem.

**Theorem 3** (Limiting Spectral Distribution). *Let $E_\pm = \left( \sqrt{\frac{c_1}{1-c_3}} \pm \sqrt{\frac{c_2}{1-c_3}} \right)^2$. As $n_1, n_2 \to +\infty$, the matrix $\frac{1}{\varsigma^2} \bar{\mathbf{T}}^\top \bar{\mathbf{T}}$ has a limiting spectral distribution $\mu$ explicitly given by*

$$\mu_{MP}(\mathrm{d}x) = \left[ 1 - \frac{c_1}{c_2} \right]^+ \delta_x(\mathrm{d}x) + \frac{1}{2\pi \frac{c_2}{1-c_3} x} \sqrt{[x - E_-]^+ [E_+ - x]^+} \mathrm{d}x.$$

Similarly to the previous spiked models, the rank-one information $\beta_T \beta_M \boldsymbol{x} \boldsymbol{y}^\top$ induces the presence of an isolated eigenvalue in the spectrum of $\frac{1}{\varsigma^2} \bar{\mathbf{T}}^\top \bar{\mathbf{T}}$. The following theorem characterize its behavior and that of its corresponding eigenvector.

**Theorem 4** (Spike Behavior). *If $\left( \frac{\beta_T \beta_M}{\varsigma} \right)^4 > \frac{c_1 c_2}{(1-c_3)^2}$, then the spectrum of $\frac{1}{\varsigma^2} \bar{\mathbf{T}}^\top \bar{\mathbf{T}}$ exhibits an isolated eigenvalue asymptotically located in $\frac{\varsigma^2}{\beta_T^2 \beta_M^2} \left( \frac{\beta_T^2 \beta_M^2}{\varsigma^2} + \frac{c_1}{1-c_3} \right) \left( \frac{\beta_T^2 \beta_M^2}{\varsigma^2} + \frac{c_2}{1-c_3} \right)$ almost surely. Moreover, in this case, the corresponding eigenvector $\hat{\mathbf{u}}$ is aligned with the signal $\boldsymbol{y}$,*

$$\left| \hat{\mathbf{u}}^\top \boldsymbol{y} \right|^2 \xrightarrow[n_1, n_2, n_3 \to +\infty]{} 1 - \frac{\varsigma^2}{\beta_T^2 \beta_M^2} \frac{c_2}{1-c_3} \frac{\frac{\beta_T^2 \beta_M^2}{\varsigma^2} + \frac{c_1}{1-c_3}}{\frac{\beta_T^2 \beta_M^2}{\varsigma^2} + \frac{c_2}{1-c_3}} \qquad \textit{almost surely.}$$

For more details on standard spiked matrix models, see Couillet & Liao (2022, §2.5).

## 4 PERFORMANCE GAPS IN MULTI-VIEW CLUSTERING

We shall now illustrate our results in the context of multi-view clustering. As explained in the introduction, we consider the observation of a tensor $\mathbf{X} \in \mathbb{R}^{p \times n \times m}$ following the nested matrix-tensor model,

$$\mathbf{X} = \left( \boldsymbol{\mu} \bar{\boldsymbol{y}}^\top + \mathbf{Z} \right) \otimes \boldsymbol{h} + \mathbf{W} \qquad \text{with} \qquad \begin{cases} Z_{i,j} \overset{\text{i.i.d.}}{\sim} \mathcal{N}\left( 0, \frac{1}{p+n} \right) \\ W_{i,j,k} \overset{\text{i.i.d.}}{\sim} \mathcal{N}\left( 0, \frac{1}{p+n+m} \right) \end{cases}. \qquad (5)$$

The two cluster centers are $\pm\boldsymbol{\mu}$ and $\bar{y}_i = \pm\frac{1}{\sqrt{n}}$ depending on the class of the $i$-th individual. The third vector $\boldsymbol{h}$ encodes the variances along the different views of $\boldsymbol{\mu} \bar{\boldsymbol{y}}^\top + \mathbf{Z}$. The clustering is performed by estimating the class labels with the dominant left singular vector $\hat{\mathbf{y}}$ of $\mathbf{X}^{(2)}$. It is thus a direct application of the results of Section 3.1, where $(\|\boldsymbol{\mu}\|, \|\boldsymbol{h}\|)$ plays the role of $(\beta_M, \beta_T)$. In fact, the behavior of the alignment $\left| \hat{\mathbf{y}}^\top \boldsymbol{y} \right|^2$ given by Theorem 2 can be further precised with the following theorem.

**Theorem 5** (Performance of Multi-View Spectral Clustering). *Let $(c_p, c_n, c_m) = \frac{(p,n,m)}{p+n+m}$, $\rho = \|\boldsymbol{h}\|^2 \frac{p+n+m}{\sqrt{pnm}}$ and*

$$\zeta = 1 - \frac{1}{\|\boldsymbol{\mu}\|^2 \left( \frac{c_n}{1-c_m} + \|\boldsymbol{\mu}\|^2 \right)} \left[ \left( \frac{\|\boldsymbol{\mu}\|^2}{\rho \left( \frac{c_n}{1-c_m} + \|\boldsymbol{\mu}\|^2 \right)} \right)^2 + \frac{c_n}{1-c_m} \left( \frac{c_p}{1-c_m} + \|\boldsymbol{\mu}\|^2 \right) \right].$$

*Then, $\sqrt{\frac{n}{1-\zeta}} (\hat{\mathbf{y}}_j - \sqrt{\zeta} \bar{y}_j) \xrightarrow[n \to +\infty]{\mathcal{D}} \mathcal{N}(0,1)$ for all $j \in \{1, \ldots, n\}$, i.e., $\hat{\mathbf{y}}_j$ approximately follows $\mathcal{N}\left( \sqrt{\zeta} \bar{y}_j, \frac{1-\zeta}{n} \right)$. Therefore, the clustering accuracy of the estimator $\hat{\mathbf{y}}$ converges almost surely to $\Phi\left( \sqrt{\frac{\zeta}{1-\zeta}} \right)$ where $\Phi : x \mapsto \frac{1}{\sqrt{2\pi}} \int_{-\infty}^x e^{-\frac{t^2}{2}} \mathrm{d}t$ is the standard Gaussian cumulative distribution function.*

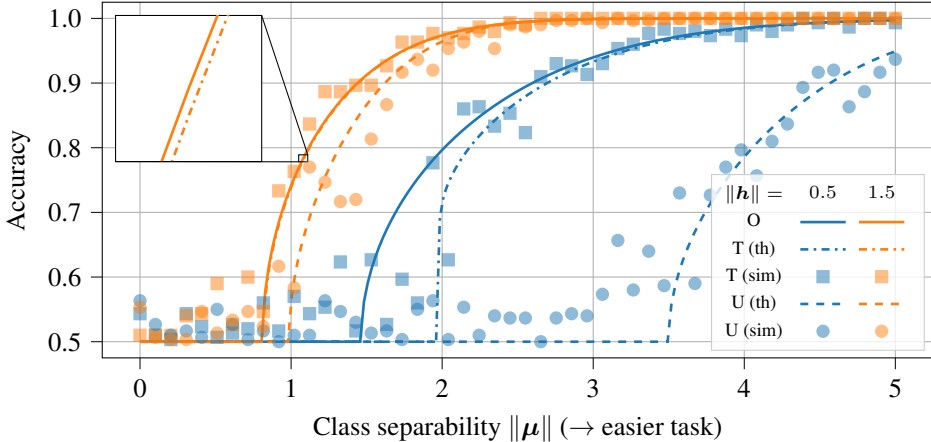

Figure 3: Empirical versus theoretical **multi-view clustering performance** with parameters $(p, n, m) = (150, 300, 60)$, varying $\|\boldsymbol{\mu}\|$ and two values of $\|\boldsymbol{h}\|$ : 0.5 in blue and 1.5 in orange. The solid curve (O) is an optimistic upper bound given by Theorem 4, as it can be reached when the variances along each view are perfectly known. The dash-dotted curve (T) is the performance achieved with a rank-one approximation of $\mathbf{X}$ (Seddik et al., 2023a). The dashed curve (U) is the performance predicted by Theorem 5 with the unfolding approach.

*Proof.* See Appendix D for a sketch. □

Figure 3 compares the performances of the unfolding approach predicted by Theorem 5 with that of the "tensor approach" (Seddik et al., 2023a) which performs clustering with a rank-one approximation of $\mathbf{X}$. Moreover, an optimistic upper bound on the best achievable performance, given by the solid curve, can be derived from Theorem 4. Empirical accuracies are computed for both approaches and show a good match between theory and simulation results. It appears that the unfolding approach has a later phase transition and a lower performance than the tensor approach. This was expected since they do not have the same non-trivial regime ($\Theta(n_T^{1/4})$ against $\Theta(1)$). As $\|\boldsymbol{h}\|$ increases, the performance gap between both approaches reduces. The performance of the tensor approach rapidly comes very close to the upper bound: the two curves almost coincide for $\|\boldsymbol{h}\| = 1.5$.

These results show the superiority of the tensor approach in terms of accuracy of the multi-view spectral clustering. In particular, by contrast with the unfolding-based estimator, the tensor approach has near-optimal performance, as quantified by Theorem 4. Nevertheless, when considering "not too hard" problems (i.e., for which $\|\boldsymbol{\mu}\|$ and $\|\boldsymbol{h}\|$ are not too close to the phase transition threshold), the performances of both methods are close and the unfolding approach may be more interesting given its ease of implementation and lower computational cost.

## 5 CONCLUSION AND PERSPECTIVES

Under the nested matrix-tensor model, we have precisely quantified the multi-view clustering performance achievable by the unfolding method and compared it with a previously studied tensor approach (Seddik et al., 2023a). This analysis has showed the theoretical superiority of the latter in terms of clustering accuracy. In particular, the tensor approach can, in principle, recover the signal at a $\Theta(1)$ signal-to-noise ratio, while the matrix approach needs this ratio to diverge as $n_T^{1/4}$. However, the tensor approach is based on an NP-hard formulation, and no efficient algorithm capable of succeeding at a $\Theta(1)$ signal-to-noise ratio is currently known. In practice, for a sufficiently large ratio, one may combine these approaches by initializing a tensor rank-one approximation algorithm (such as power iteration) with the estimate given by the unfolding method. Overall, this work advances our understanding of tensor data processing through rigorous theoretical results. Finally, a natural question for future work is how to extend our results to more general view functions $f_k$.

ACKNOWLEDGMENTS

J. H. de M. Goulart was supported by the ANR LabEx CIMI (ANR-11-LABX-0040) within the French program "Investissements d'Avenir.".

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

# A  Unfolding Along the Third Mode

For the sake of completeness, we study in this section the recovery of $z$ from the third unfolding of $\mathbf{T}$. Following the model in equation 2, the expression of $\mathbf{T}^{(3)}$ is

$$\mathbf{T}^{(3)} = \beta_T z m^\top + \frac{1}{\sqrt{n_T}} \mathbf{W}^{(3)} \qquad (6)$$

where $m = [\mathbf{M}_{1,:} \quad \ldots \quad \mathbf{M}_{n_1,:}]^\top \in \mathbb{R}^{n_1 n_2}$.

This unfolding has the peculiarity that the rank-one perturbation $\beta_T z m^\top$ mixes signal (the vector $z$) and noise (contained in $\mathbf{M}$). Still, as for the previous unfoldings, the dominant left singular vector of $\mathbf{T}^{(3)}$ remains a natural estimator of $z$ and we study the asymptotic spectral properties of $\mathbf{T}^{(3)} \mathbf{T}^{(3)\top}$. Because of the long shape of $\mathbf{T}^{(3)}$ (one dimension grows faster than the other), the spectrum of $\mathbf{T}^{(3)} \mathbf{T}^{(3)\top}$ diverges in the same way as that of $\mathbf{T}^{(2)} \mathbf{T}^{(2)\top}$. Therefore, we must proceed to a similar rescaling. The following theorem states that, after proper rescaling, the distribution of eigenvalues of $\mathbf{T}^{(3)} \mathbf{T}^{(3)\top}$ approaches the semi-circle distribution.

**Theorem 6** (Limiting Spectral Distribution). *As $n_1, n_2, n_3 \to +\infty$, the limiting spectral distribution of the centered-and-scaled matrix $\frac{n_T}{\sqrt{n_1 n_2 n_3}} \mathbf{T}^{(3)} \mathbf{T}^{(3)\top} - \frac{n_3 + n_1 n_2}{\sqrt{n_1 n_2 n_3}} I_{n_3}$ converges weakly to a semi-circle distribution on $[-2, 2]$,*

$$\mu_{SC}(\mathrm{d}x) = \frac{1}{2\pi} \sqrt{[4 - x^2]^+}.$$

*Proof.* See Appendix E.  □

The ESD and LSD of $\mathbf{T}^{(3)} \mathbf{T}^{(3)\top}$ are plotted in the right panel of Figure 1. The result of Theorem 6 is not surprising: the "non-trivial" shape of the LSD of $\mathbf{T}^{(2)} \mathbf{T}^{(2)\top}$ (Theorem 1) is due to the presence of a "signal-noise" $\frac{\beta_T}{\sqrt{n_M}} \mathbf{Z}^\top (I_{n_1} \boxtimes z)^\top$ in the expression of $\mathbf{T}^{(2)}$ but, when $\beta_T$ is set to 0, we have observed that the LSD of $\mathbf{T}^{(2)} \mathbf{T}^{(2)\top}$ is simply a semi-circle. This is coherent with the case that interest us here: in $\mathbf{T}^{(3)}$, the "signal-noise" is restrained to the rank-one perturbation and therefore does not impact the LSD of $\mathbf{T}^{(3)} \mathbf{T}^{(3)\top}$, which is then a semi-circle.

Because of the rank-one perturbation $\beta_T z m^\top$, the spectrum of $\mathbf{T}^{(3)} \mathbf{T}^{(3)\top}$ exhibits an isolated eigenvalue which can be observed in the right panel of Figure 1. Our next step is to characterize the behavior of this spike eigenvalue and the correlation with $z$ of its corresponding eigenvector. Before introducing the formal result in Theorem 7, let us have a close look at the expression of $\mathbf{T}^{(3)} \mathbf{T}^{(3)\top}$ to understand, with hand-waving arguments, what should be the non-trivial regime in this case.

$$\mathbf{T}^{(3)} \mathbf{T}^{(3)\top} = \beta_T^2 \|\mathbf{M}\|_F^2 zz^\top + \frac{\beta_T}{\sqrt{n_T}} \left( z m^\top \mathbf{W}^{(3)\top} + \mathbf{W}^{(3)} m z^\top \right) + \frac{1}{n_T} \mathbf{W}^{(3)} \mathbf{W}^{(3)\top}$$

Starting from the right, the term $\frac{1}{n_T} \mathbf{W}^{(3)} \mathbf{W}^{(3)\top}$ is already understood thanks to Theorem 6 and yields a semi-circle as limiting spectral distribution. The crossed-terms in the middle have zero mean and are expected to vanish. On the left, remains the rank-one term $zz^\top$ weighted by $\beta_T^2 \|\mathbf{M}\|_F^2$, which is a random quantity because of the noise $\mathbf{Z}$ in $\mathbf{M}$. However, the quantity $\|\mathbf{M}\|_F^2$ is expected to rapidly concentrate around its mean $\frac{n_1 n_2}{n_M} + \beta_M^2$. Hence, guessing from the results on the previous unfoldings, we would need the quantity $\beta_T^2 \left( \frac{n_1 n_2}{n_M} + \beta_M^2 \right) \frac{n_T}{\sqrt{n_1 n_2 n_3}}$ to converge to a fixed positive value denoted $\varrho$. Indeed, this is precisely what is found when this analysis is rigorously carried out (see Appendix E), meaning that $\beta_T = \Theta(n_T^{-1/4})$. In other words, if $\beta_T$ is a constant ($\beta_T = \Theta(1)$), we are above the non-trivial regime and therefore should expect (asymptotically) exact recovery. This is because the strength of the signal is "boosted" by $\|\mathbf{M}\|_F^2 = \Theta(n_T)$.

*Remark.* $\varrho$ is defined as the limit of $\beta_T^2 \left( \frac{n_1 n_2}{n_M} + \beta_M^2 \right) \frac{n_T}{\sqrt{n_1 n_2 n_3}}$ which is the same as that of $\beta_T^2 \frac{n_1 n_2}{n_M} \frac{n_T}{\sqrt{n_1 n_2 n_3}}$ since $\beta_M = \Theta(1)$. However, we keep the $\beta_M^2$ term in the definition of $\rho_T$ as it yields better predictions in our simulations.

**Theorem 7** (Spike Behavior). *If* $\varrho = \lim \frac{\beta_T^2 n_T}{\sqrt{n_1 n_2 n_3}} \left( \frac{n_1 n_2}{n_M} + \beta_M^2 \right) > 1$, *then the centered-and-scaled matrix* $\frac{n_T}{\sqrt{n_1 n_2 n_3}} \mathbf{T}^{(3)} \mathbf{T}^{(3)\top} - \frac{n_3 + n_1 n_2}{\sqrt{n_1 n_2 n_3}} \boldsymbol{I}_{n_3}$ *has an isolated eigenvalue asymptotically located in* $\varrho + \frac{1}{\varrho}$ *almost surely. Furthermore, in this case, the alignment between the corresponding eigenvector* $\hat{\mathbf{z}}$ *and the true signal* $\boldsymbol{z}$ *converges to* $1 - \frac{1}{\varrho^2}$ *almost surely, i.e.,*

$$\left| \hat{\mathbf{z}}^\top \boldsymbol{z} \right|^2 \xrightarrow[n_1, n_2, n_3 \to +\infty]{} 1 - \frac{1}{\varrho^2} \qquad almost\ surely.$$

*Proof.* See Appendix F. □

Once the quantity $\varrho$ is defined, we recognize in Theorem 7 the same results as that of the spiked Wigner model (Benaych-Georges & Nadakuditi, 2011).

*Remark.* In practice, we work with *large but finite* tensors. Hence, it makes no sense to say that $\beta_T = \Theta(n_T^{1/4})$ or $\beta_T = \Theta(n_T^{-1/4})$. In fact, the characterization of the "non-trivial" regime is only important here to reveal the relevant quantities, i.e., $\rho_T$ and $\varrho$, which we will use in practice without worrying whether $\beta_T$ is in the right regime or not.

## B    PROOF OF THEOREM 1

Denote $\mathbf{Q}(s) = \left( \mathbf{T}^{(2)} \mathbf{T}^{(2)\top} - s \boldsymbol{I}_{n_2} \right)^{-1}$ the resolvent of $\mathbf{T}^{(2)} \mathbf{T}^{(2)\top}$ defined for all $s \in \mathbb{C} \setminus \mathrm{sp}\, \mathbf{T}^{(2)} \mathbf{T}^{(2)\top}$.

### B.1    COMPUTATIONS WITH STEIN'S LEMMA

Before delving into the analysis of $\mathbf{Q}$, we will derive a few useful results thanks to Stein's lemma (Lemma 1). They are gathered in the following Proposition 2.

**Proposition 2.**

$$\mathbb{E} \left[ \mathbf{W}^{(2)} \mathbf{T}^{(2)\top} \mathbf{Q} \right] = \frac{n_1 n_3}{\sqrt{n_T}} \mathbb{E} \left[ \mathbf{Q} \right] - \frac{1}{\sqrt{n_T}} \mathbb{E} \left[ (n_2 + 1) \mathbf{Q} + s \left( \mathbf{Q} \operatorname{Tr} \mathbf{Q} + \mathbf{Q}^2 \right) \right] \tag{7}$$

$$\mathbb{E} \left[ \mathbf{M}^\top \left( \boldsymbol{I}_{n_1} \boxtimes \boldsymbol{z} \right)^\top \mathbf{T}^{(2)\top} \left( \mathbf{Q} + \frac{1}{n_T} \left( \mathbf{Q} \operatorname{Tr} \mathbf{Q} + \mathbf{Q}^2 \right) \right) \right] = \beta_T \mathbb{E} \left[ \mathbf{M}^\top \mathbf{M} \mathbf{Q} \right] \tag{8}$$

$$\mathbb{E} \left[ \mathbf{Z}^\top \mathbf{M} \mathbf{Q} \right] = \frac{1}{\sqrt{n_M}} \mathbb{E} \left[ (n_1 - n_2 - 1) \mathbf{Q} - \left( s - \frac{n_1 n_3}{n_T} \right) \left( \mathbf{Q} \operatorname{Tr} \mathbf{Q} + \mathbf{Q}^2 \right) \right] \tag{9}$$

$$- \frac{s}{n_T \sqrt{n_M}} \mathbb{E} \left[ 4 \mathbf{Q}^3 + 2 \mathbf{Q}^2 \operatorname{Tr} \mathbf{Q} + \mathbf{Q} \operatorname{Tr} \mathbf{Q}^2 + \mathbf{Q} \operatorname{Tr}^2 \mathbf{Q} \right]$$

$$- \frac{1}{n_T \sqrt{n_M}} \mathbb{E} \left[ (n_2 + 2) \mathbf{Q} \operatorname{Tr} \mathbf{Q} + (n_2 + 4) \mathbf{Q}^2 \right]$$

$$\mathbb{E} \left[ \boldsymbol{y} \boldsymbol{x}^\top \mathbf{Z} \mathbf{Q} \right] = -\frac{\beta_T}{\sqrt{n_M}} \mathbb{E} \left[ \boldsymbol{y} \left( \boldsymbol{x} \boxtimes \boldsymbol{z} \right)^\top \mathbf{T}^{(2)\top} \left( \mathbf{Q} \operatorname{Tr} \mathbf{Q} + \mathbf{Q}^2 \right) \right] \tag{10}$$

$$\beta_T \mathbb{E} \left[ \boldsymbol{y} \boldsymbol{x}^\top \mathbf{M} \mathbf{Q} \right] = \mathbb{E} \left[ \boldsymbol{y} \left( \boldsymbol{x} \boxtimes \boldsymbol{z} \right)^\top \mathbf{T}^{(2)\top} \left( \mathbf{Q} + \frac{1}{n_T} \left( \mathbf{Q} \operatorname{Tr} \mathbf{Q} + \mathbf{Q}^2 \right) \right) \right] \tag{11}$$

In order to prove these results, we will need the following expressions for the derivatives of $\mathbf{Q}$.

**Proposition 3.**

$$\frac{\partial \mathbf{Q}_{a,b}}{\partial \mathbf{W}_{c,d}^{(2)}} = -\frac{1}{\sqrt{n_T}} \left( \mathbf{Q}_{a,c} \left[ \mathbf{T}^{(2)\top} \mathbf{Q} \right]_{d,b} + \mathbf{Q}_{c,b} \left[ \mathbf{T}^{(2)\top} \mathbf{Q} \right]_{d,a} \right) \tag{12}$$

$$\frac{\partial \mathbf{Q}_{a,b}}{\partial \mathbf{Z}_{c,d}} = -\frac{\beta_T}{\sqrt{n_M}} \left( \mathbf{Q}_{a,d} \left[ \mathbf{Q} \mathbf{T}^{(2)} \left( \boldsymbol{I}_{n_1} \boxtimes \boldsymbol{z} \right) \right]_{b,c} + \mathbf{Q}_{d,b} \left[ \mathbf{Q} \mathbf{T}^{(2)} \left( \boldsymbol{I}_{n_1} \boxtimes \boldsymbol{z} \right) \right]_{a,c} \right) \tag{13}$$

*Proof.* Since $\partial \mathbf{Q} = -\mathbf{Q} \partial \left( \mathbf{T}^{(2)} \mathbf{T}^{(2)\top} \right) \mathbf{Q}$,

$$\frac{\partial \mathbf{Q}_{a,b}}{\partial \mathsf{W}_{i,j,k}} = -\sum_{e=1}^{n_2} \sum_{f=1}^{n_1} \sum_{g=1}^{n_3} \sum_{h=1}^{n_2} \mathbf{Q}_{a,e} \left( \frac{\partial \mathsf{T}_{f,e,g}}{\partial \mathsf{W}_{i,j,k}} \mathsf{T}_{f,h,g} + \mathsf{T}_{f,e,g} \frac{\partial \mathsf{T}_{f,h,g}}{\partial \mathsf{W}_{i,j,k}} \right) \mathbf{Q}_{h,b}$$

$$= -\frac{1}{\sqrt{n_T}} \left( \sum_{h=1}^{n_2} \mathbf{Q}_{a,j} \mathsf{T}_{i,h,k} \mathbf{Q}_{h,b} + \sum_{e=1}^{n_2} \mathbf{Q}_{a,e} \mathsf{T}_{i,e,k} \mathbf{Q}_{j,b} \right)$$

$$\frac{\partial \mathbf{Q}_{a,b}}{\partial \mathsf{W}_{i,j,k}} = -\frac{1}{\sqrt{n_T}} \left( \mathbf{Q}_{a,j} \left[ \mathbf{T}^{(2)\top} \mathbf{Q} \right]_{[i,k],b} + \mathbf{Q}_{j,b} \left[ \mathbf{T}^{(2)\top} \mathbf{Q} \right]_{[i,k],a} \right).$$

Likewise,

$$\frac{\partial \mathbf{Q}_{a,b}}{\partial \mathbf{Z}_{c,d}} = -\sum_{e=1}^{n_2} \sum_{f=1}^{n_1} \sum_{g=1}^{n_3} \sum_{h=1}^{n_2} \mathbf{Q}_{a,e} \left( \frac{\partial \mathsf{T}_{f,e,g}}{\partial \mathbf{Z}_{c,d}} \mathsf{T}_{f,h,g} + \mathsf{T}_{f,e,g} \frac{\partial \mathsf{T}_{f,h,g}}{\partial \mathbf{Z}_{c,d}} \right) \mathbf{Q}_{h,b}$$

$$= -\frac{\beta_T}{\sqrt{n_M}} \left( \sum_{g=1}^{n_3} \sum_{h=1}^{n_2} \mathbf{Q}_{a,d} z_g \mathsf{T}_{c,h,g} \mathbf{Q}_{h,b} + \sum_{e=1}^{n_2} \sum_{g=1}^{n_3} \mathbf{Q}_{a,e} \mathsf{T}_{c,e,g} z_g \mathbf{Q}_{d,b} \right)$$

$$\frac{\partial \mathbf{Q}_{a,b}}{\partial \mathbf{Z}_{c,d}} = -\frac{\beta_T}{\sqrt{n_M}} \left( \mathbf{Q}_{a,d} \left[ \mathbf{Q} \mathbf{T}^{(2)} \left( \mathbf{I}_{n_1} \boxtimes \mathbf{z} \right) \right]_{b,c} + \mathbf{Q}_{d,b} \left[ \mathbf{Q} \mathbf{T}^{(2)} \left( \mathbf{I}_{n_1} \boxtimes \mathbf{z} \right) \right]_{a,c} \right).$$

$\square$

Combining Stein's lemma (Lemma 1) and Proposition 3, we can prove each expression of Proposition 2.

**Proof of equation 7**

$$\mathbb{E} \left[ \mathbf{W}^{(2)} \mathbf{T}^{(2)\top} \mathbf{Q} \right]_{i,j} = \sum_{k=1}^{n_1 n_3} \sum_{l=1}^{n_2} \mathbb{E} \left[ \mathsf{W}^{(2)}_{i,k} \mathsf{T}^{(2)}_{l,k} \mathbf{Q}_{l,j} \right]$$

$$= \sum_{k=1}^{n_1 n_3} \sum_{l=1}^{n_2} \mathbb{E} \left[ \frac{\partial \mathsf{T}^{(2)}_{l,k}}{\partial \mathsf{W}^{(2)}_{i,k}} \mathbf{Q}_{l,j} + \mathsf{T}^{(2)}_{l,k} \frac{\partial \mathbf{Q}_{l,j}}{\partial \mathsf{W}^{(2)}_{i,k}} \right]$$

$$= \frac{n_1 n_3}{\sqrt{n_T}} \mathbb{E} \left[ \mathbf{Q} \right]_{i,j} - \frac{1}{\sqrt{n_T}} \sum_{k=1}^{n_1 n_3} \sum_{l=1}^{n_2} \mathbb{E} \left[ \mathsf{T}^{(2)}_{l,k} \left( \mathbf{Q}_{l,i} \left[ \mathbf{T}^{(2)\top} \mathbf{Q} \right]_{k,j} + \mathbf{Q}_{i,j} \left[ \mathbf{T}^{(2)\top} \mathbf{Q} \right]_{k,l} \right) \right]$$

$$= \frac{n_1 n_3}{\sqrt{n_T}} \mathbb{E} \left[ \mathbf{Q} \right]_{i,j} - \frac{1}{\sqrt{n_T}} \mathbb{E} \left[ \mathbf{Q} \mathbf{T}^{(2)} \mathbf{T}^{(2)\top} \mathbf{Q} + \mathbf{Q} \operatorname{Tr} \left( \mathbf{T}^{(2)} \mathbf{T}^{(2)\top} \mathbf{Q} \right) \right]_{i,j}$$

$$= \frac{n_1 n_3}{\sqrt{n_T}} \mathbb{E} \left[ \mathbf{Q} \right]_{i,j} - \frac{1}{\sqrt{n_T}} \mathbb{E} \left[ \left( n_2 + 1 \right) \mathbf{Q} + s \left( \mathbf{Q} \operatorname{Tr} \mathbf{Q} + \mathbf{Q}^2 \right) \right]_{i,j}$$

where the last equality comes from $\mathbf{T}^{(2)} \mathbf{T}^{(2)\top} \mathbf{Q} = \mathbf{I}_{n_2} + s\mathbf{Q}$.

**Proof of equation 8**

$$
\mathbb{E}\left[\mathbf{M}^{\top}\left(\boldsymbol{I}_{n_1}\boxtimes \boldsymbol{z}\right)^{\top}\mathbf{T}^{(2)\top}\mathbf{Q}\right]_{i,j}
$$

$$
=\beta_T\mathbb{E}\left[\mathbf{M}^{\top}\mathbf{M}\mathbf{Q}\right]_{i,j}+\frac{1}{\sqrt{n_T}}\sum_{k=1}^{n_1 n_3}\sum_{l=1}^{n_2}\mathbb{E}\left[\left[\mathbf{M}^{\top}\left(\boldsymbol{I}_{n_1}\boxtimes \boldsymbol{z}\right)^{\top}\right]_{i,k}\mathbf{W}_{l,k}^{(2)}\mathbf{Q}_{l,j}\right]
$$

$$
=\beta_T\mathbb{E}\left[\mathbf{M}^{\top}\mathbf{M}\mathbf{Q}\right]_{i,j}+\frac{1}{\sqrt{n_T}}\sum_{k=1}^{n_1 n_3}\sum_{l=1}^{n_2}\mathbb{E}\left[\left[\mathbf{M}^{\top}\left(\boldsymbol{I}_{n_1}\boxtimes \boldsymbol{z}\right)^{\top}\right]_{i,k}\frac{\partial \mathbf{Q}_{l,j}}{\partial \mathbf{W}_{l,k}^{(2)}}\right]
$$

$$
=\beta_T\mathbb{E}\left[\mathbf{M}^{\top}\mathbf{M}\mathbf{Q}\right]_{i,j}-\frac{1}{n_T}\sum_{k=1}^{n_1 n_3}\sum_{l=1}^{n_2}\mathbb{E}\left[\left[\mathbf{M}^{\top}\left(\boldsymbol{I}_{n_1}\boxtimes \boldsymbol{z}\right)^{\top}\right]_{i,k}\mathbf{Q}_{l,l}\left[\mathbf{T}^{(2)\top}\mathbf{Q}\right]_{k,j}\right]
$$

$$
-\frac{1}{n_T}\sum_{k=1}^{n_1 n_3}\sum_{l=1}^{n_2}\mathbb{E}\left[\left[\mathbf{M}^{\top}\left(\boldsymbol{I}_{n_1}\boxtimes \boldsymbol{z}\right)^{\top}\right]_{i,k}\mathbf{Q}_{l,j}\left[\mathbf{T}^{(2)\top}\mathbf{Q}\right]_{k,l}\right]
$$

$$
=\beta_T\mathbb{E}\left[\mathbf{M}^{\top}\mathbf{M}\mathbf{Q}\right]_{i,j}-\frac{1}{n_T}\mathbb{E}\left[\mathbf{M}^{\top}\left(\boldsymbol{I}_{n_1}\boxtimes \boldsymbol{z}\right)^{\top}\mathbf{T}^{(2)\top}\left(\mathbf{Q}\operatorname{Tr}\mathbf{Q}+\mathbf{Q}^2\right)\right]_{i,j}.
$$

**Proof of equation 9**

$$
\mathbb{E}\left[\mathbf{Z}^{\top}\mathbf{M}\mathbf{Q}\right]_{i,j}=\sum_{k=1}^{n_1}\sum_{l=1}^{n_2}\mathbb{E}\left[\mathbf{Z}_{k,i}\mathbf{M}_{k,l}\mathbf{Q}_{l,j}\right]
$$

$$
=\frac{n_1}{\sqrt{n_M}}\mathbb{E}\left[\mathbf{Q}\right]_{i,j}+\sum_{k=1}^{n_1}\sum_{l=1}^{n_2}\mathbb{E}\left[\mathbf{M}_{k,l}\frac{\partial \mathbf{Q}_{l,j}}{\partial \mathbf{Z}_{k,i}}\right]
$$

$$
=\frac{n_1}{\sqrt{n_M}}\mathbb{E}\left[\mathbf{Q}\right]_{i,j}
$$

$$
-\frac{\beta_T}{\sqrt{n_M}}\sum_{k=1}^{n_1}\sum_{l=1}^{n_2}\mathbb{E}\left[\mathbf{M}_{k,l}\left(\mathbf{Q}_{l,i}\left[\mathbf{Q}\mathbf{T}^{(2)}\left(\boldsymbol{I}_{n_1}\boxtimes \boldsymbol{z}\right)\right]_{j,k}+\mathbf{Q}_{i,j}\left[\mathbf{Q}\mathbf{T}^{(2)}\left(\boldsymbol{I}_{n_1}\boxtimes \boldsymbol{z}\right)\right]_{l,k}\right)\right]
$$

$$
=\frac{n_1}{\sqrt{n_M}}\mathbb{E}\left[\mathbf{Q}\right]_{i,j}-\frac{\beta_T}{\sqrt{n_M}}\mathbb{E}\left[\mathbf{Q}\mathbf{M}^{\top}\left(\boldsymbol{I}_{n_1}\boxtimes \boldsymbol{z}\right)^{\top}\mathbf{T}^{(2)\top}\mathbf{Q}+\mathbf{Q}\operatorname{Tr}\left(\mathbf{M}^{\top}\left(\boldsymbol{I}_{n_1}\boxtimes \boldsymbol{z}\right)^{\top}\mathbf{T}^{(2)\top}\mathbf{Q}\right)\right]_{i,j}
$$

and, since $\beta_T\mathbf{M}^{\top}\left(\boldsymbol{I}_{n_1}\boxtimes \boldsymbol{z}\right)^{\top}=\mathbf{T}^{(2)}-\frac{1}{\sqrt{n_T}}\mathbf{W}^{(2)}$, with the relation $\mathbf{T}^{(2)}\mathbf{T}^{(2)\top}\mathbf{Q}=s\mathbf{Q}+\boldsymbol{I}_{n_2}$ we have,

$$
\mathbb{E}\left[\mathbf{Z}^{\top}\mathbf{M}\mathbf{Q}\right]_{i,j}=\frac{n_1}{\sqrt{n_M}}\mathbb{E}\left[\mathbf{Q}\right]_{i,j}-\frac{1}{\sqrt{n_M}}\mathbb{E}\left[s\mathbf{Q}^2+\mathbf{Q}+s\mathbf{Q}\operatorname{Tr}\mathbf{Q}+n_2\mathbf{Q}\right]_{i,j}
$$

$$
+\frac{1}{\sqrt{n_M n_T}}\mathbb{E}\left[\mathbf{Q}\mathbf{W}^{(2)}\mathbf{T}^{(2)\top}\mathbf{Q}+\mathbf{Q}\operatorname{Tr}\left(\mathbf{W}^{(2)}\mathbf{T}^{(2)\top}\mathbf{Q}\right)\right]_{i,j}
$$

$$
=\frac{n_1}{\sqrt{n_M}}\mathbb{E}\left[\mathbf{Q}\right]_{i,j}-\frac{1}{\sqrt{n_M}}\mathbb{E}\left[\left(n_2+1\right)\mathbf{Q}+s\left(\mathbf{Q}\operatorname{Tr}\mathbf{Q}+\mathbf{Q}^2\right)\right]_{i,j}
$$

$$
+\frac{1}{\sqrt{n_M n_T}}\sum_{u=1}^{n_2}\sum_{v=1}^{n_1 n_3}\sum_{w=1}^{n_2}\mathbb{E}\left[\frac{\partial \mathbf{Q}_{i,u}}{\partial \mathbf{W}_{u,v}^{(2)}}\mathbf{T}_{w,v}^{(2)}\mathbf{Q}_{w,j}+\mathbf{Q}_{i,u}\frac{\partial \mathbf{T}_{w,v}^{(2)}}{\partial \mathbf{W}_{u,v}^{(2)}}\mathbf{Q}_{w,j}+\mathbf{Q}_{i,u}\mathbf{T}_{w,v}^{(2)}\frac{\partial \mathbf{Q}_{w,j}}{\partial \mathbf{W}_{u,v}^{(2)}}\right]
$$

$$
+\frac{1}{\sqrt{n_M n_T}}\sum_{u=1}^{n_2}\sum_{v=1}^{n_1 n_3}\sum_{w=1}^{n_2}\mathbb{E}\left[\frac{\partial \mathbf{Q}_{i,j}}{\partial \mathbf{W}_{u,v}^{(2)}}\mathbf{T}_{w,v}^{(2)}\mathbf{Q}_{w,u}+\mathbf{Q}_{i,j}\frac{\partial \mathbf{T}_{w,v}^{(2)}}{\partial \mathbf{W}_{u,v}^{(2)}}\mathbf{Q}_{w,u}+\mathbf{Q}_{i,j}\mathbf{T}_{w,v}^{(2)}\frac{\partial \mathbf{Q}_{w,u}}{\partial \mathbf{W}_{u,v}^{(2)}}\right]
$$

where we have used Stein's lemma (Lemma 1) again. Hence,

$$
\mathbb{E}\left[\mathbf{Z}^\top \mathbf{M}\mathbf{Q}\right]_{i,j}
$$

$$
= \frac{n_1}{\sqrt{n_M}}\mathbb{E}\left[\mathbf{Q}\right]_{i,j} - \frac{1}{\sqrt{n_M}}\mathbb{E}\left[(n_2+1)\,\mathbf{Q} + s\left(\mathbf{Q}\operatorname{Tr}\mathbf{Q} + \mathbf{Q}^2\right)\right]_{i,j} + \frac{n_1 n_3}{n_T \sqrt{n_M}}\mathbb{E}\left[\mathbf{Q}^2 + \mathbf{Q}\operatorname{Tr}\mathbf{Q}\right]_{i,j}
$$

$$
- \frac{1}{n_T \sqrt{n_M}}\sum_{u=1}^{n_2}\sum_{v=1}^{n_1 n_3}\sum_{w=1}^{n_2}\mathbb{E}\left[\left(\mathrm{Q}_{i,u}\left[\mathbf{T}^{(2)\top}\mathbf{Q}\right]_{v,u} + \mathrm{Q}_{u,u}\left[\mathbf{T}^{(2)\top}\mathbf{Q}\right]_{v,i}\right)\mathrm{T}^{(2)}_{w,v}\mathrm{Q}_{w,j}\right]
$$

$$
- \frac{1}{n_T \sqrt{n_M}}\sum_{u=1}^{n_2}\sum_{v=1}^{n_1 n_3}\sum_{w=1}^{n_2}\mathbb{E}\left[\mathrm{Q}_{i,u}\mathrm{T}^{(2)}_{w,v}\left(\mathrm{Q}_{w,u}\left[\mathbf{T}^{(2)\top}\mathbf{Q}\right]_{v,j} + \mathrm{Q}_{u,j}\left[\mathbf{T}^{(2)\top}\mathbf{Q}\right]_{v,w}\right)\right]
$$

$$
- \frac{1}{n_T \sqrt{n_M}}\sum_{u=1}^{n_2}\sum_{v=1}^{n_1 n_3}\sum_{w=1}^{n_2}\mathbb{E}\left[\left(\mathrm{Q}_{i,u}\left[\mathbf{T}^{(2)\top}\mathbf{Q}\right]_{v,j} + \mathrm{Q}_{u,j}\left[\mathbf{T}^{(2)\top}\mathbf{Q}\right]_{v,i}\right)\mathrm{T}^{(2)}_{w,v}\mathrm{Q}_{w,u}\right]
$$

$$
- \frac{1}{n_T \sqrt{n_M}}\sum_{u=1}^{n_2}\sum_{v=1}^{n_1 n_3}\sum_{w=1}^{n_2}\mathbb{E}\left[\mathrm{Q}_{i,j}\mathrm{T}^{(2)}_{w,v}\left(\mathrm{Q}_{w,u}\left[\mathbf{T}^{(2)\top}\mathbf{Q}\right]_{v,u} + \mathrm{Q}_{u,u}\left[\mathbf{T}^{(2)\top}\mathbf{Q}\right]_{v,w}\right)\right]
$$

which bunches up into

$$
\mathbb{E}\left[\mathbf{Z}^\top \mathbf{M}\mathbf{Q}\right]_{i,j}
$$

$$
= \frac{n_1}{\sqrt{n_M}}\mathbb{E}\left[\mathbf{Q}\right]_{i,j} - \frac{1}{\sqrt{n_M}}\mathbb{E}\left[(n_2+1)\,\mathbf{Q} + \left(s - \frac{n_1 n_3}{n_T}\right)\left(\mathbf{Q}\operatorname{Tr}\mathbf{Q} + \mathbf{Q}^2\right)\right]_{i,j}
$$

$$
- \frac{1}{n_T \sqrt{n_M}}\mathbb{E}\left[\mathbf{Q}^2\mathbf{T}^{(2)}\mathbf{T}^{(2)\top}\mathbf{Q} + \mathbf{Q}\mathbf{T}^{(2)}\mathbf{T}^{(2)\top}\mathbf{Q}\operatorname{Tr}\mathbf{Q}\right]_{i,j}
$$

$$
- \frac{1}{n_T \sqrt{n_M}}\mathbb{E}\left[\mathbf{Q}^2\mathbf{T}^{(2)}\mathbf{T}^{(2)\top}\mathbf{Q} + \mathbf{Q}^2\operatorname{Tr}\left(\mathbf{T}^{(2)}\mathbf{T}^{(2)\top}\mathbf{Q}\right)\right]_{i,j}
$$

$$
- \frac{1}{n_T \sqrt{n_M}}\mathbb{E}\left[\mathbf{Q}^2\mathbf{T}^{(2)}\mathbf{T}^{(2)\top}\mathbf{Q} + \mathbf{Q}\mathbf{T}^{(2)}\mathbf{T}^{(2)\top}\mathbf{Q}^2\right]_{i,j}
$$

$$
- \frac{1}{n_T \sqrt{n_M}}\mathbb{E}\left[\mathbf{Q}\operatorname{Tr}\left(\mathbf{Q}\mathbf{T}^{(2)}\mathbf{T}^{(2)\top}\mathbf{Q}\right) + \mathbf{Q}\operatorname{Tr}\mathbf{Q}\operatorname{Tr}\left(\mathbf{T}^{(2)}\mathbf{T}^{(2)\top}\mathbf{Q}\right)\right]_{i,j}
$$

and, using again the relation $\mathbf{T}^{(2)}\mathbf{T}^{(2)\top}\mathbf{Q} = s\mathbf{Q} + \boldsymbol{I}_{n_2}$, we get,

$$
\mathbb{E}\left[\mathbf{Z}^\top \mathbf{M}\mathbf{Q}\right]
$$

$$
= \frac{n_1}{\sqrt{n_M}}\mathbb{E}\left[\mathbf{Q}\right] - \frac{1}{\sqrt{n_M}}\mathbb{E}\left[(n_2+1)\,\mathbf{Q} + \left(s - \frac{n_1 n_3}{n_T}\right)\left(\mathbf{Q}\operatorname{Tr}\mathbf{Q} + \mathbf{Q}^2\right)\right]
$$

$$
- \frac{1}{n_T \sqrt{n_M}}\mathbb{E}\left[s\mathbf{Q}^3 + \mathbf{Q}^2 + s\mathbf{Q}^2\operatorname{Tr}\mathbf{Q} + \mathbf{Q}\operatorname{Tr}\mathbf{Q} + s\mathbf{Q}^3 + \mathbf{Q}^2 + s\mathbf{Q}^2\operatorname{Tr}\mathbf{Q} + n_2\mathbf{Q}^2\right]
$$

$$
- \frac{1}{n_T \sqrt{n_M}}\mathbb{E}\left[s\mathbf{Q}^3 + \mathbf{Q}^2 + s\mathbf{Q}^3 + \mathbf{Q}^2 + s\mathbf{Q}\operatorname{Tr}\mathbf{Q}^2 + \mathbf{Q}\operatorname{Tr}\mathbf{Q} + s\mathbf{Q}\operatorname{Tr}^2\mathbf{Q} + n_2\mathbf{Q}\operatorname{Tr}\mathbf{Q}\right]
$$

$$
= \frac{n_1}{\sqrt{n_M}}\mathbb{E}\left[\mathbf{Q}\right] - \frac{1}{\sqrt{n_M}}\mathbb{E}\left[(n_2+1)\,\mathbf{Q} + \left(s - \frac{n_1 n_3}{n_T}\right)\left(\mathbf{Q}\operatorname{Tr}\mathbf{Q} + \mathbf{Q}^2\right)\right]
$$

$$
- \frac{1}{n_T \sqrt{n_M}}\mathbb{E}\left[s\left(4\mathbf{Q}^3 + 2\mathbf{Q}^2\operatorname{Tr}\mathbf{Q} + \mathbf{Q}\operatorname{Tr}\mathbf{Q}^2 + \mathbf{Q}\operatorname{Tr}^2\mathbf{Q}\right)\right]
$$

$$
- \frac{1}{n_T \sqrt{n_M}}\mathbb{E}\left[(n_2+4)\,\mathbf{Q}^2 + (n_2+2)\,\mathbf{Q}\operatorname{Tr}\mathbf{Q}\right].
$$

**Proof of equation 10**

$$\mathbb{E}\left[\boldsymbol{y}\boldsymbol{x}^\top \mathbf{Z}\mathbf{Q}\right]_{i,j} = \sum_{k=1}^{n_1}\sum_{l=1}^{n_2}\mathbb{E}\left[\left[\boldsymbol{y}\boldsymbol{x}^\top\right]_{i,k} \mathbf{Z}_{k,l}\mathbf{Q}_{l,j}\right]$$

$$= \sum_{k=1}^{n_1}\sum_{l=1}^{n_2}\mathbb{E}\left[\left[\boldsymbol{y}\boldsymbol{x}^\top\right]_{i,k} \frac{\partial \mathbf{Q}_{l,j}}{\partial \mathbf{Z}_{k,l}}\right]$$

$$= -\frac{\beta_T}{\sqrt{n_M}}\sum_{k=1}^{n_1}\sum_{l=1}^{n_2}\mathbb{E}\left[\left[\boldsymbol{y}\boldsymbol{x}^\top\right]_{i,k} \left(\mathbf{Q}_{l,l}\left[\mathbf{Q}\mathbf{T}^{(2)}\left(\boldsymbol{I}_{n_1} \boxtimes \boldsymbol{z}\right)\right]_{j,k} + \mathbf{Q}_{l,j}\left[\mathbf{Q}\mathbf{T}^{(2)}\left(\boldsymbol{I}_{n_1} \boxtimes \boldsymbol{z}\right)\right]_{l,k}\right)\right]$$

$$= -\frac{\beta_T}{\sqrt{n_M}}\mathbb{E}\left[\boldsymbol{y}\left(\boldsymbol{x} \boxtimes \boldsymbol{z}\right)^\top \mathbf{T}^{(2)\top}\left(\mathbf{Q}\operatorname{Tr}\mathbf{Q} + \mathbf{Q}^2\right)\right]_{i,j}.$$

**Proof of equation 11**

$$\beta_T\mathbb{E}\left[\boldsymbol{y}\boldsymbol{x}^\top \mathbf{M}\mathbf{Q}\right] = \beta_T\mathbb{E}\left[\boldsymbol{y}\left(\boldsymbol{x} \boxtimes \boldsymbol{z}\right)^\top\left(\boldsymbol{I}_{n_1} \boxtimes \boldsymbol{z}\right)\mathbf{M}\mathbf{Q}\right]$$

$$= \mathbb{E}\left[\boldsymbol{y}\left(\boldsymbol{x} \boxtimes \boldsymbol{z}\right)^\top\left(\mathbf{T}^{(2)} - \frac{1}{\sqrt{n_T}}\mathbf{W}^{(2)}\right)^\top \mathbf{Q}\right]$$

and,

$$\mathbb{E}\left[\boldsymbol{y}\left(\boldsymbol{x} \boxtimes \boldsymbol{z}\right)^\top \mathbf{W}^{(2)\top}\mathbf{Q}\right]_{i,j} = \sum_{k=1}^{n_1 n_3}\sum_{l=1}^{n_2}\mathbb{E}\left[\left[\boldsymbol{y}\left(\boldsymbol{x} \boxtimes \boldsymbol{z}\right)^\top\right]_{i,k} \mathbf{W}^{(2)}_{l,k}\mathbf{Q}_{l,j}\right]$$

$$= \sum_{k=1}^{n_1 n_3}\sum_{l=1}^{n_2}\mathbb{E}\left[\left[\boldsymbol{y}\left(\boldsymbol{x} \boxtimes \boldsymbol{z}\right)^\top\right]_{i,k} \frac{\partial \mathbf{Q}_{l,j}}{\partial \mathbf{W}^{(2)}_{l,k}}\right]$$

$$= -\frac{1}{\sqrt{n_T}}\sum_{k=1}^{n_1 n_3}\sum_{l=1}^{n_2}\mathbb{E}\left[\left[\boldsymbol{y}\left(\boldsymbol{x} \boxtimes \boldsymbol{z}\right)^\top\right]_{i,k} \left(\mathbf{Q}_{l,l}\left[\mathbf{T}^{(2)\top}\mathbf{Q}\right]_{k,j} + \mathbf{Q}_{l,j}\left[\mathbf{T}^{(2)\top}\mathbf{Q}\right]_{k,l}\right)\right]$$

$$= -\frac{1}{\sqrt{n_T}}\mathbb{E}\left[\boldsymbol{y}\left(\boldsymbol{x} \boxtimes \boldsymbol{z}\right)^\top \mathbf{T}^{(2)\top}\left(\mathbf{Q}\operatorname{Tr}\mathbf{Q} + \mathbf{Q}^2\right)\right]_{i,j}.$$

### B.2 DETERMINISTIC EQUIVALENT

Since $\mathbf{Q}^{-1}\mathbf{Q} = \boldsymbol{I}_{n_2}$, we have $\mathbf{T}^{(2)}\mathbf{T}^{(2)\top}\mathbf{Q} - s\mathbf{Q} = \boldsymbol{I}_{n_2}$. Hence, using equation 3 and taking the expectation,

$$\beta_T\mathbb{E}\left[\mathbf{M}^\top\left(\boldsymbol{I}_{n_1} \boxtimes \boldsymbol{z}\right)^\top \mathbf{T}^{(2)\top}\mathbf{Q}\right] + \frac{1}{\sqrt{n_T}}\mathbb{E}\left[\mathbf{W}^{(2)}\mathbf{T}^{(2)\top}\mathbf{Q}\right] - s\mathbb{E}\left[\mathbf{Q}\right] = \boldsymbol{I}_{n_2}$$

and, injecting equation 7, this yields

$$\beta_T\mathbb{E}\left[\mathbf{M}^\top\left(\boldsymbol{I}_{n_1} \boxtimes \boldsymbol{z}\right)^\top \mathbf{T}^{(2)\top}\mathbf{Q}\right]$$
$$+ \frac{n_1 n_3}{n_T}\mathbb{E}\left[\mathbf{Q}\right] - \frac{1}{n_T}\mathbb{E}\left[\left(n_2 + 1\right)\mathbf{Q} + s\left(\mathbf{Q}\operatorname{Tr}\mathbf{Q} + \mathbf{Q}^2\right)\right] - s\mathbb{E}\left[\mathbf{Q}\right] = \boldsymbol{I}_{n_2}.$$

Here, we must be careful that, due to the $n_2 \times n_1 n_3$ rectangular shape of $\mathbf{T}^{(2)}$, the spectrum of $\mathbf{T}^{(2)}\mathbf{T}^{(2)\top}$ diverges in the limit $n_1, n_2, n_3 \to +\infty$. In order to bypass this obstacle, we shall perform a change of variable $(s, \mathbf{Q}) \curvearrowright (\tilde{s}, \tilde{\mathbf{Q}})$ which will become clearer after rearranging the terms:

$$s\frac{n_2}{n_T}\mathbb{E}\left[\frac{\operatorname{Tr}\mathbf{Q}}{n_2}\mathbf{Q}\right] + \left(s + \frac{n_2 - n_1 n_3}{n_T}\right)\mathbb{E}\left[\mathbf{Q}\right] + \boldsymbol{I}_{n_2}$$
$$= \beta_T\mathbb{E}\left[\mathbf{M}^\top\left(\boldsymbol{I}_{n_1} \boxtimes \boldsymbol{z}\right)^\top \mathbf{T}^{(2)\top}\mathbf{Q}\right] - \frac{1}{n_T}\mathbb{E}\left[\mathbf{Q} + s\mathbf{Q}^2\right].$$

Let $\tilde{s} = \frac{n_T s - (n_2 + n_1 n_3)}{\sqrt{n_1 n_2 n_3}}$ and $\tilde{\mathbf{Q}}(\tilde{s}) = \left( \frac{n_T \mathbf{T}^{(2)} \mathbf{T}^{(2)\top} - (n_2 + n_1 n_3) \, \boldsymbol{I}_{n_2}}{\sqrt{n_1 n_2 n_3}} - \tilde{s} \boldsymbol{I}_{n_2} \right)^{-1}$. With this rescaling, note that we have $\mathbf{Q}(s) = \frac{n_T}{\sqrt{n_1 n_2 n_3}} \tilde{\mathbf{Q}}(\tilde{s})$, and the previous equation becomes

$$\left( \frac{n_T}{\sqrt{n_1 n_2 n_3}} \tilde{s} + \frac{n_T}{n_1 n_3} + \frac{n_T}{n_2} \right) \frac{n_2}{n_T} \mathbb{E}\left[ \frac{\mathrm{Tr}\, \tilde{\mathbf{Q}}}{n_2} \tilde{\mathbf{Q}} \right] + \left( \tilde{s} + \frac{2 n_2}{\sqrt{n_1 n_2 n_3}} \right) \mathbb{E}\left[ \tilde{\mathbf{Q}} \right] + \boldsymbol{I}_{n_2}$$

$$= \frac{\beta_T n_T}{\sqrt{n_1 n_2 n_3}} \mathbb{E}\left[ \mathbf{M}^\top \left( \boldsymbol{I}_{n_1} \boxtimes \boldsymbol{z} \right)^\top \mathbf{T}^{(2)\top} \tilde{\mathbf{Q}} \right] - \frac{1}{\sqrt{n_1 n_2 n_3}} \mathbb{E}\left[ \tilde{\mathbf{Q}} + \left( \tilde{s} + \frac{n_2 + n_1 n_3}{\sqrt{n_1 n_2 n_3}} \right) \tilde{\mathbf{Q}}^2 \right]$$

and we no longer have any diverging terms. Keeping only the dominant terms, we can now proceed to a matrix-equivalent formula[7]

$$\frac{\mathrm{Tr}\, \tilde{\mathbf{Q}}}{n_2} \tilde{\mathbf{Q}} + \tilde{s} \tilde{\mathbf{Q}} + \boldsymbol{I}_{n_2} \longleftrightarrow \frac{\beta_T n_T}{\sqrt{n_1 n_2 n_3}} \mathbf{M}^\top \left( \boldsymbol{I}_{n_1} \boxtimes \boldsymbol{z} \right)^\top \mathbf{T}^{(2)\top} \tilde{\mathbf{Q}}.$$

*Remark.* This previous step is justified by the use of standard concentration arguments such as Poincaré-Nash inequality (Chen, 1982; Ledoux, 2001) and Borel-Cantelli lemma (Billingsley, 2012), following, e.g., the lines of Couillet & Liao (2022, §2.2.2). This reasoning is applied in what follows whenever we move from an equality with expectations to an equality with matrix equivalents.

Applying the same rescaling to equation 8 yields

$$\frac{\beta_T n_T}{\sqrt{n_1 n_2 n_3}} \mathbf{M}^\top \left( \boldsymbol{I}_{n_1} \boxtimes \boldsymbol{z} \right)^\top \mathbf{T}^{(2)\top} \tilde{\mathbf{Q}} \longleftrightarrow \frac{\beta_T^2 n_T}{\sqrt{n_1 n_2 n_3}} \mathbf{M}^\top \mathbf{M} \tilde{\mathbf{Q}}.$$

*Remark.* We see, here, that we must chose $\beta_T = \Theta(n_T^{1/4})$ in order to place ourselves in the non-trivial regime where both the noise and the signal are $\Theta(1)$. This is different from the estimation of $\boldsymbol{y}$ with a rank-one approximation of $\mathbf{T}$, where $\beta_T$ could be kept $\Theta(1)$ (Seddik et al., 2023a) (but at the cost of an NP-hard computation (Hillar & Lim, 2013)).

With this in mind, let us denote $\rho_T = \lim \frac{\beta_T^2 n_T}{\sqrt{n_1 n_2 n_3}}$ and $\tilde{m} = \lim \frac{\mathrm{Tr}\, \tilde{\mathbf{Q}}}{n_2}$ the Stieltjes transform of the limiting spectral distribution of the centered-and-scaled matrix $\frac{n_T}{\sqrt{n_1 n_2 n_3}} \mathbf{T}^{(2)} \mathbf{T}^{(2)\top} - \frac{n_2 + n_1 n_3}{\sqrt{n_1 n_2 n_3}} \boldsymbol{I}_{n_2}$. We have,

$$\tilde{m}(\tilde{s}) \tilde{\mathbf{Q}} + \tilde{s} \tilde{\mathbf{Q}} + \boldsymbol{I}_{n_2} \longleftrightarrow \rho_T \mathbf{M}^\top \mathbf{M} \tilde{\mathbf{Q}}.$$

In order to handle the term $\mathbf{M}^\top \mathbf{M} \tilde{\mathbf{Q}}$, observe that

$$\mathbb{E}\left[ \mathbf{M}^\top \mathbf{M} \tilde{\mathbf{Q}} \right] = \beta_M \mathbb{E}\left[ \boldsymbol{y} \boldsymbol{x}^\top \mathbf{M} \tilde{\mathbf{Q}} \right] + \frac{1}{\sqrt{n_M}} \mathbb{E}\left[ \mathbf{Z}^\top \mathbf{M} \tilde{\mathbf{Q}} \right]$$

and, after rescaling, the only non-vanishing terms in equation 9 yield

$$\frac{1}{\sqrt{n_M}} \mathbf{Z}^\top \mathbf{M} \tilde{\mathbf{Q}} \longleftrightarrow \left( \frac{n_1 - n_2}{n_M} - \tilde{s} \frac{n_2}{n_M} \frac{\mathrm{Tr}\, \tilde{\mathbf{Q}}}{n_2} - \frac{n_2}{n_M} \left( \frac{\mathrm{Tr}\, \tilde{\mathbf{Q}}}{n_2} \right)^2 \right) \tilde{\mathbf{Q}}$$

Moreover, from equation 10, we have

$$\beta_M \boldsymbol{y} \boldsymbol{x}^\top \mathbf{M} \tilde{\mathbf{Q}} \longleftrightarrow \beta_M^2 \boldsymbol{y} \boldsymbol{y}^\top \tilde{\mathbf{Q}} - \frac{\beta_T \beta_M n_T n_2}{n_M \sqrt{n_1 n_2 n_3}} \frac{\mathrm{Tr}\, \tilde{\mathbf{Q}}}{n_2} \boldsymbol{y} \left( \boldsymbol{x} \boxtimes \boldsymbol{z} \right)^\top \mathbf{T}^{(2)\top} \tilde{\mathbf{Q}}$$

but, from equation 11, we also know that

$$\boldsymbol{y} \boldsymbol{x}^\top \mathbf{M} \tilde{\mathbf{Q}} \longleftrightarrow \frac{1}{\beta_T} \boldsymbol{y} \left( \boldsymbol{x} \boxtimes \boldsymbol{z} \right)^\top \mathbf{T}^{(2)\top} \tilde{\mathbf{Q}}.$$

---

[7]We say that two (sequences of) matrices $\mathbf{A}, \mathbf{B} \in \mathbb{R}^{n \times n}$ are equivalent, and we write $\mathbf{A} \longleftrightarrow \mathbf{B}$, if, for all $\boldsymbol{M} \in \mathbb{R}^{n \times n}$ and $\boldsymbol{u}, \boldsymbol{v} \in \mathbb{R}^n$ of unit norms (respectively, operator and Euclidean),

$$\frac{1}{n} \mathrm{Tr}\, \boldsymbol{M} \left( \mathbf{A} - \mathbf{B} \right) \xrightarrow[n \to +\infty]{\text{a.s.}} 0, \quad \boldsymbol{u}^\top \left( \mathbf{A} - \mathbf{B} \right) \boldsymbol{v} \xrightarrow[n \to +\infty]{\text{a.s.}} 0, \quad \left\| \mathbb{E}\left[ \mathbf{A} - \mathbf{B} \right] \right\| \xrightarrow[n \to +\infty]{\text{a.s.}} 0.$$

Hence,

$$\frac{1}{\beta_T}\left(\beta_M + \frac{\beta_T^2 \beta_M n_T n_2}{n_M \sqrt{n_1 n_2 n_3}} \frac{\operatorname{Tr} \tilde{\mathbf{Q}}}{n_2}\right) \boldsymbol{y} \left(\boldsymbol{x} \boxtimes \boldsymbol{z}\right)^\top \mathbf{T}^{(2)\top} \tilde{\mathbf{Q}} \longleftrightarrow \beta_M^2 \boldsymbol{y}\boldsymbol{y}^\top \tilde{\mathbf{Q}}$$

and

$$\beta_M \boldsymbol{y}\boldsymbol{x}^\top \mathbf{M}\tilde{\mathbf{Q}} \longleftrightarrow \beta_M^2 \left(1 - \frac{\frac{\beta_T^2 \beta_M n_T n_2}{n_M \sqrt{n_1 n_2 n_3}} \frac{\operatorname{Tr} \tilde{\mathbf{Q}}}{n_2}}{\beta_M + \frac{\beta_T^2 \beta_M n_T n_2}{n_M \sqrt{n_1 n_2 n_3}} \frac{\operatorname{Tr} \tilde{\mathbf{Q}}}{n_2}}\right) \boldsymbol{y}\boldsymbol{y}^\top \tilde{\mathbf{Q}}.$$

Therefore, we have the following matrix equivalent for $\mathbf{M}^\top \mathbf{M}\tilde{\mathbf{Q}}$,

$$\mathbf{M}^\top \mathbf{M}\tilde{\mathbf{Q}} \longleftrightarrow \frac{\beta_M^2}{1 + \frac{\beta_T^2 n_T n_2}{n_M \sqrt{n_1 n_2 n_3}} \frac{\operatorname{Tr} \tilde{\mathbf{Q}}}{n_2}} \boldsymbol{y}\boldsymbol{y}^\top \tilde{\mathbf{Q}}$$

$$+ \left(\frac{n_1}{n_M} - \frac{n_2}{n_M}\left(1 + \tilde{s}\frac{\operatorname{Tr} \tilde{\mathbf{Q}}}{n_2} + \left(\frac{\operatorname{Tr} \tilde{\mathbf{Q}}}{n_2}\right)^2\right)\right) \tilde{\mathbf{Q}}.$$

Ultimately, we can define $\bar{\mathbf{Q}}$, the deterministic equivalent of $\tilde{\mathbf{Q}}$ such that,

$$\tilde{m}(\tilde{s})\bar{\mathbf{Q}} + \tilde{s}\bar{\mathbf{Q}} + \boldsymbol{I}_{n_2}$$

$$= \rho_T \left[\frac{\beta_M^2}{1 + \frac{\rho_T c_2}{1 - c_3}\tilde{m}(\tilde{s})} \boldsymbol{y}\boldsymbol{y}^\top + \left(\frac{c_1}{1 - c_3} - \frac{c_2}{1 - c_3}\left(1 + \tilde{s}\tilde{m}(\tilde{s}) + \tilde{m}^2(\tilde{s})\right)\right) \boldsymbol{I}_{n_2}\right]\bar{\mathbf{Q}}$$

or, equivalently,

$$\left[\frac{\rho_T c_2}{1 - c_3}\tilde{m}^2(\tilde{s}) + \left(1 + \tilde{s}\frac{\rho_T c_2}{1 - c_3}\right)\tilde{m}(\tilde{s}) + \tilde{s} + \frac{\rho_T(c_2 - c_1)}{1 - c_3}\right]\bar{\mathbf{Q}} + \boldsymbol{I}_{n_2} = \frac{\rho_T \beta_M^2}{1 + \frac{\rho_T c_2}{1 - c_3}\tilde{m}(\tilde{s})} \boldsymbol{y}\boldsymbol{y}^\top \bar{\mathbf{Q}}.$$

(14)

### B.3 LIMITING SPECTRAL DISTRIBUTION

According to the previous deterministic equivalent, the Stieltjes transform of the limiting spectral distribution of $\frac{n_T}{\sqrt{n_1 n_2 n_3}}\mathbf{T}^{(2)}\mathbf{T}^{(2)\top} - \frac{n_2 + n_1 n_3}{\sqrt{n_1 n_2 n_3}}\boldsymbol{I}_{n_2}$ is solution to

$$\frac{\rho_T c_2}{1 - c_3}\tilde{m}^3(\tilde{s}) + \left(1 + \tilde{s}\frac{\rho_T c_2}{1 - c_3}\right)\tilde{m}^2(\tilde{s}) + \left(\tilde{s} + \frac{\rho_T(c_2 - c_1)}{1 - c_3}\right)\tilde{m}(\tilde{s}) + 1 = 0$$

$$\iff \left(\frac{\rho_T c_2}{1 - c_3}\tilde{m}(\tilde{s}) + 1\right)\left(\tilde{m}^2(\tilde{s}) + \tilde{s}\tilde{m}(\tilde{s}) + 1\right) - \frac{\rho_T c_1}{1 - c_3}\tilde{m}(\tilde{s}) = 0. \quad (15)$$

## C PROOF OF THEOREM 2

### C.1 ISOLATED EIGENVALUE

The asymptotic position $\tilde{\xi}$ of the isolated eigenvalue (when it exists) is a singular point of $\bar{\mathbf{Q}}$. From equation 14, it is therefore solution to

$$\frac{\rho_T c_2}{1 - c_3}\tilde{m}^2(\tilde{\xi}) + \left(1 + \tilde{\xi}\frac{\rho_T c_2}{1 - c_3}\right)\tilde{m}(\tilde{\xi}) + \tilde{\xi} + \frac{\rho_T(c_2 - c_1)}{1 - c_3} = \frac{\rho_T \beta_M^2}{1 + \frac{\rho_T c_2}{1 - c_3}\tilde{m}(\tilde{\xi})}$$

$$\iff \left(1 + \frac{\rho_T c_2}{1 - c_3}\tilde{m}(\tilde{\xi})\right)\left[\frac{\rho_T c_2}{1 - c_3}\left(\tilde{m}^2(\tilde{\xi}) + \tilde{\xi}\tilde{m}(\tilde{\xi}) + 1\right) + \tilde{m}(\tilde{\xi}) + \tilde{\xi} - \frac{\rho_T c_1}{1 - c_3}\right] = \rho_T \beta_M^2.$$

Yet, $\left(\frac{\rho_T c_2}{1-c_3}\tilde{m}(\tilde{s}) + 1\right)\left(\tilde{m}^2(\tilde{s}) + \tilde{s}\tilde{m}(\tilde{s}) + 1\right) = \frac{\rho_T c_1}{1-c_3}\tilde{m}(\tilde{s})$ (from equation 15). Thus,

$$\tilde{m}(\tilde{\xi}) + \tilde{\xi} - \frac{\rho_T c_1}{1-c_3} + \frac{\rho_T c_2}{1-c_3}\tilde{m}(\tilde{\xi})\left(\tilde{m}(\tilde{\xi}) + \tilde{\xi}\right) = \rho_T \beta_M^2$$

$$\iff \underbrace{\left(1 + \frac{\rho_T c_2}{1-c_3}\tilde{m}(\tilde{\xi})\right)\left(\tilde{m}(\tilde{\xi}) + \tilde{\xi}\right)}_{=\frac{\rho_T c_1}{1-c_3} - \frac{\rho_T c_2}{1-c_3} - \frac{1}{\tilde{m}(\tilde{\xi})} \quad \text{from equation 15 again}} - \frac{\rho_T c_1}{1-c_3} = \rho_T \beta_M^2$$

$$\iff \tilde{m}(\tilde{\xi}) = \frac{-1}{\frac{\rho_T c_2}{1-c_3} + \rho_T \beta_M^2}.$$

Injecting this relation into equation 15 yields

$$\tilde{\xi} = \frac{\rho_T}{\beta_M^2}\left(\frac{c_1}{1-c_3} + \beta_M^2\right)\left(\frac{c_2}{1-c_3} + \beta_M^2\right) + \frac{1}{\rho_T\left(\frac{c_2}{1-c_3} + \beta_M^2\right)}. \tag{16}$$

## C.2 EIGENVECTOR ALIGNMENT

The eigendecomposition of $\tilde{\mathbf{Q}}(\tilde{s})$ is given by $\sum_{i=1}^n \frac{\mathbf{u}_i\mathbf{u}_i^\top}{\tilde{\lambda}_i - \tilde{s}}$ where $(\tilde{\lambda}_i, \mathbf{u}_i)_{1\leqslant i\leqslant n}$ are the eigenvalue-eigenvector pairs of $\frac{n_T \mathbf{T}^{(2)}\mathbf{T}^{(2)\top} - (n_2 + n_1 n_3)\mathbf{I}_{n_2}}{\sqrt{n_1 n_2 n_3}}$. Hence, thanks to Cauchy's integral formula, we have

$$\left|\boldsymbol{y}^\top \hat{\mathbf{y}}\right|^2 = -\frac{1}{2i\pi}\oint_{\tilde{\gamma}} \boldsymbol{y}^\top \tilde{\mathbf{Q}}(\tilde{s})\boldsymbol{y}\,\mathrm{d}\tilde{s}$$

where $\tilde{\gamma}$ is a positively-oriented complex contour circling around the isolated eigenvalue only. The asymptotic value $\zeta$ of $\left|\boldsymbol{y}^\top \hat{\mathbf{y}}\right|^2$ can then be computed with the deterministic equivalent defined in equation 14,

$$\zeta = -\frac{1}{2i\pi}\oint_{\tilde{\gamma}} \boldsymbol{y}^\top \bar{\mathbf{Q}}(\tilde{s})\boldsymbol{y}\,\mathrm{d}\tilde{s}.$$

Using residue calculus, we shall compute,

$$\zeta = \lim_{\tilde{s}\to\tilde{\xi}} \frac{\tilde{s} - \tilde{\xi}}{\frac{\rho_T c_2}{1-c_3}\tilde{m}^2(\tilde{s}) + \left(1 + \tilde{s}\frac{\rho_T c_2}{1-c_3}\right)\tilde{m}(\tilde{s}) + \tilde{s} + \frac{\rho_T(c_2-c_1)}{1-c_3} - \frac{\rho_T \beta_M^2}{1 + \frac{\rho_T c_2}{1-c_3}\tilde{m}(\tilde{s})}}.$$

The limit can be expressed using L'Hôpital's rule,

$$\zeta = \frac{1}{\frac{\mathrm{d}}{\mathrm{d}\tilde{s}}\left[\frac{\rho_T c_2}{1-c_3}\tilde{m}^2(\tilde{s}) + \left(1 + \tilde{s}\frac{\rho_T c_2}{1-c_3}\right)\tilde{m}(\tilde{s}) + \tilde{s} + \frac{\rho_T(c_2-c_1)}{1-c_3} - \frac{\rho_T \beta_M^2}{1 + \frac{\rho_T c_2}{1-c_3}\tilde{m}(\tilde{s})}\right]_{\tilde{s}=\tilde{\xi}}}$$

$$= \left[\left(2\frac{\rho_T c_2}{1-c_3}\tilde{m}(\tilde{\xi}) + 1 + \tilde{\xi}\frac{\rho_T c_2}{1-c_3} + \frac{\rho_T \beta_M^2 \frac{\rho_T c_2}{1-c_3}}{\left(1 + \frac{\rho_T c_2}{1-c_3}\tilde{m}(\tilde{\xi})\right)^2}\right)\tilde{m}'(\tilde{\xi}) + \frac{\rho_T c_2}{1-c_3}\tilde{m}(\tilde{\xi}) + 1\right]^{-1}$$

We already know that $\tilde{m}(\tilde{\xi}) = \frac{-1}{\frac{\rho_T c_2}{1-c_3} + \rho_T \beta_M^2}$. Let us then differentiate equation 15,

$$\left(3\frac{\rho_T c_2}{1-c_3}\tilde{m}^2(\tilde{s}) + 2\left(1 + \tilde{s}\frac{\rho_T c_2}{1-c_3}\right)\tilde{m}(\tilde{s}) + \tilde{s} + \frac{\rho_T(c_2-c_1)}{1-c_3}\right)\tilde{m}'(\tilde{s})$$

$$+ \frac{\rho_T c_2}{1-c_3}\tilde{m}^2(\tilde{s}) + \tilde{m}(\tilde{s}) = 0.$$

Since $\left(\frac{\rho_T c_2}{1-c_3}\tilde{m}^2(\tilde{s}) + \left(1 + \tilde{s}\frac{\rho_T c_2}{1-c_3}\right)\tilde{m}(\tilde{s}) + \tilde{s} + \frac{\rho_T(c_2-c_1)}{1-c_3}\right)\tilde{m}(\tilde{s}) + 1 = 0$, we have,

$$\left(2\frac{\rho_T c_2}{1-c_3}\tilde{m}^2(\tilde{s}) + \left(1 + \tilde{s}\frac{\rho_T c_2}{1-c_3}\right)\tilde{m}(\tilde{s}) - \frac{1}{\tilde{m}(\tilde{s})}\right)\tilde{m}'(\tilde{s}) + \frac{\rho_T c_2}{1-c_3}\tilde{m}^2(\tilde{s}) + \tilde{m}(\tilde{s}) = 0$$

$$\iff \left(2\frac{\rho_T c_2}{1-c_3}\tilde{m}(\tilde{s}) + 1 + \tilde{s}\frac{\rho_T c_2}{1-c_3} - \frac{1}{\tilde{m}^2(\tilde{s})}\right)\tilde{m}'(\tilde{s}) + \frac{\rho_T c_2}{1-c_3}\tilde{m}(\tilde{s}) + 1 = 0.$$

From this last equality, we see that

$$\zeta = \left[ \left( \frac{1}{m^2(\tilde\xi)} + \frac{\rho_T \beta_M^2 \frac{\rho_T c_2}{1-c_3}}{\left(1 + \frac{\rho_T c_2}{1-c_3}\tilde m(\tilde\xi)\right)^2} \right) \tilde m'(\tilde\xi) \right]^{-1} = -\rho_T \beta_M^2 \frac{\tilde m^3(\tilde\xi)}{\tilde m'(\tilde\xi)}.$$

Moreover,

$$\tilde m'(\tilde\xi) = -\frac{1 + \frac{\rho_T c_2}{1-c_3}\tilde m(\tilde\xi)}{2\frac{\rho_T c_2}{1-c_3}\tilde m(\tilde\xi) + 1 + \tilde\xi\frac{\rho_T c_2}{1-c_3} - \frac{1}{\tilde m^2(\tilde\xi)}}$$

$$= \frac{-\frac{\frac{\rho_T \beta_M^2}{\rho_T c_2}{1-c_3}+\rho_T\beta_M^2}}{-2\frac{\frac{\rho_T c_2}{1-c_3}}{\frac{\rho_T c_2}{1-c_3}+\rho_T\beta_M^2} + 1 + \tilde\xi\frac{\rho_T c_2}{1-c_3} - \left(\frac{\rho_T c_2}{1-c_3}+\rho_T\beta_M^2\right)^2}$$

$$= \frac{\rho_T\beta_M^2}{\frac{\rho_T c_2}{1-c_3} - \rho_T\beta_M^2 - \tilde\xi\frac{\rho_T c_2}{1-c_3}\left(\frac{\rho_T c_2}{1-c_3}+\rho_T\beta_M^2\right) + \left(\frac{\rho_T c_2}{1-c_3}+\rho_T\beta_M^2\right)^3}$$

$$\tilde m'(\tilde\xi) = \frac{\rho_T\beta_M^2}{-\rho_T\beta_M^2 - \frac{1}{\rho_T\beta_M^2}\frac{\rho_T c_2}{1-c_3}\left(\frac{\rho_T c_1}{1-c_3}+\rho_T\beta_M^2\right)\left(\frac{\rho_T c_2}{1-c_3}+\rho_T\beta_M^2\right)^2 + \left(\frac{\rho_T c_2}{1-c_3}+\rho_T\beta_M^2\right)^3}.$$

Hence,

$$\zeta = \frac{-\rho_T\beta_M^2 - \frac{1}{\rho_T\beta_M^2}\frac{\rho_T c_2}{1-c_3}\left(\frac{\rho_T c_1}{1-c_3}+\rho_T\beta_M^2\right)\left(\frac{\rho_T c_2}{1-c_3}+\rho_T\beta_M^2\right)^2 + \left(\frac{\rho_T c_2}{1-c_3}+\rho_T\beta_M^2\right)^3}{\left(\frac{\rho_T c_2}{1-c_3}+\rho_T\beta_M^2\right)^3}$$

$$= 1 - \frac{\rho_T\beta_M^2}{\left(\frac{\rho_T c_2}{1-c_3}+\rho_T\beta_M^2\right)^3} - \frac{1}{\rho_T\beta_M^2}\frac{\rho_T c_2}{1-c_3}\frac{\frac{\rho_T c_1}{1-c_3}+\rho_T\beta_M^2}{\frac{\rho_T c_2}{1-c_3}+\rho_T\beta_M^2}$$

$$\zeta = 1 - \frac{1}{\beta_M^2\left(\frac{c_2}{1-c_3}+\beta_M^2\right)}\left[\left(\frac{\beta_M^2}{\rho_T\left(\frac{c_2}{1-c_3}+\beta_M^2\right)}\right)^2 + \frac{c_2}{1-c_3}\left(\frac{c_1}{1-c_3}+\beta_M^2\right)\right]. \quad (17)$$

# D    SKETCH OF PROOF OF THEOREM 5

$\hat{\boldsymbol{y}}$ can be decomposed as $\alpha\bar{\boldsymbol{y}}+\beta\bar{\boldsymbol{y}}_\perp$ where $\bar{\boldsymbol{y}}_\perp$ is a unit norm vector orthogonal to $\bar{\boldsymbol{y}}$ and $\alpha^2+\beta^2=1$. From Theorem 2, we know that $\alpha^2 \to \zeta$ (and thus $\beta^2 \to 1-\zeta$) almost surely as $n_1, n_2, n_3 \to +\infty$.

From the rotational invariance of the model, $\bar{\boldsymbol{y}}_\perp$ is uniformly distributed on the unit sphere in the subspace orthogonal to $\bar{\boldsymbol{y}}$. Hence, any finite collection of its entries is asymptotically Gaussian (Diaconis & Freedman, 1987).

# E    PROOF OF THEOREM 6

Let $\mathbf{Q}(s) = \left(\mathbf{T}^{(3)}\mathbf{T}^{(3)\top} - s\boldsymbol{I}_{n_3}\right)^{-1}$ defined for all $s \in \mathbb{C} \setminus \mathrm{sp}\,\mathbf{T}^{(3)}\mathbf{T}^{(3)\top}$.

## E.1    PRELIMINARY RESULTS

Let us derive a few useful results for the upcoming analysis.

**Lemma 2.**

$$\frac{\partial \mathbf{Q}_{a,b}}{\partial \mathbf{W}_{c,d}^{(3)}} = -\frac{1}{\sqrt{n_T}}\left(\mathbf{Q}_{a,c}\left[\mathbf{T}^{(3)\top}\mathbf{Q}\right]_{d,b} + \mathbf{Q}_{c,b}\left[\mathbf{T}^{(3)\top}\mathbf{Q}\right]_{d,a}\right).$$

*Proof.* Since $\partial\mathbf{Q} = -\mathbf{Q}\partial\left(\mathbf{T}^{(3)}\mathbf{T}^{(3)\top}\right)\mathbf{Q}$,

$$\frac{\partial\mathbf{Q}_{a,b}}{\partial\mathrm{W}_{c,d}^{(3)}} = -\left[\mathbf{Q}\frac{\partial\mathbf{T}^{(3)}\mathbf{T}^{(3)\top}}{\partial\mathrm{W}_{c,d}^{(3)}}\mathbf{Q}\right]_{a,b}$$

$$= -\sum_{e,f,g=1}^{n_3,n_1n_2,n_3}\mathbf{Q}_{a,e}\left(\frac{\partial\mathrm{T}_{e,f}^{(3)}}{\partial\mathrm{W}_{c,d}^{(3)}}\mathrm{T}_{g,f}^{(3)} + \mathrm{T}_{e,f}^{(3)}\frac{\partial\mathrm{T}_{g,f}^{(3)}}{\partial\mathrm{W}_{c,d}^{(3)}}\right)\mathbf{Q}_{g,b}$$

$$= -\frac{1}{\sqrt{n_T}}\sum_{e,f,g=1}^{n_3,n_1n_2,n_3}\mathbf{Q}_{a,e}\left(\delta_{e,c}\delta_{f,d}\mathrm{T}_{g,f}^{(3)} + \mathrm{T}_{e,f}^{(3)}\delta_{g,c}\delta_{f,d}\right)\mathbf{Q}_{g,b}$$

$$\frac{\partial\mathbf{Q}_{a,b}}{\partial\mathrm{W}_{c,d}^{(3)}} = -\frac{1}{\sqrt{n_T}}\left(\mathbf{Q}_{a,c}\left[\mathbf{T}^{(3)\top}\mathbf{Q}\right]_{d,b} + \mathbf{Q}_{c,b}\left[\mathbf{T}^{(3)\top}\mathbf{Q}\right]_{d,a}\right).$$

$\square$

**Proposition 4.**

$$\mathbb{E}\left[\mathbf{W}^{(3)}\mathbf{T}^{(3)\top}\mathbf{Q}\right] = \frac{n_1n_2}{\sqrt{n_T}}\mathbb{E}\left[\mathbf{Q}\right] - \frac{1}{\sqrt{n_T}}\mathbb{E}\left[(n_3+1)\mathbf{Q} + s\left(\mathbf{Q}^2 + \mathbf{Q}\operatorname{Tr}\mathbf{Q}\right)\right] \quad (18)$$

$$\mathbb{E}\left[\boldsymbol{z}\boldsymbol{m}^\top\mathbf{T}^{(3)\top}\left(\mathbf{Q} + \frac{1}{n_T}\left(\mathbf{Q}\operatorname{Tr}\mathbf{Q} + \mathbf{Q}^2\right)\right)\right] = \beta_T\mathbb{E}\left[\|\mathbf{M}\|_{\mathrm{F}}^2\,\boldsymbol{z}\boldsymbol{z}^\top\mathbf{Q}\right] \quad (19)$$

*Proof.* We prove these two results with Stein's lemma (Lemma 1) and Lemma 2.

$$\mathbb{E}\left[\mathbf{W}^{(3)}\mathbf{T}^{(3)\top}\mathbf{Q}\right]_{i,j} = \sum_{k,l=1}^{n_1n_2,n_3}\mathbb{E}\left[\mathrm{W}_{i,k}^{(3)}\mathrm{T}_{l,k}^{(3)}\mathbf{Q}_{l,j}\right]$$

$$= \sum_{k,l=1}^{n_1n_2,n_3}\mathbb{E}\left[\frac{\partial\mathrm{T}_{l,k}^{(3)}}{\partial\mathrm{W}_{i,k}^{(3)}}\mathbf{Q}_{l,j} + \mathrm{T}_{l,k}^{(3)}\frac{\partial\mathbf{Q}_{l,j}}{\partial\mathrm{W}_{i,k}^{(3)}}\right]$$

$$= \sum_{k,l=1}^{n_1n_2,n_3}\mathbb{E}\left[\frac{\delta_{l,i}\delta_{k,k}}{\sqrt{n_T}}\mathbf{Q}_{l,j}\right]$$

$$\quad - \frac{1}{\sqrt{n_T}}\sum_{k,l=1}^{n_1n_2,n_3}\mathbb{E}\left[\mathrm{T}_{l,k}^{(3)}\left(\mathbf{Q}_{l,i}\left[\mathbf{T}^{(3)\top}\mathbf{Q}\right]_{k,j} + \mathbf{Q}_{i,j}\left[\mathbf{T}^{(3)\top}\mathbf{Q}\right]_{k,l}\right)\right]$$

$$= \frac{n_1n_2}{\sqrt{n_T}}\mathbb{E}\left[\mathbf{Q}\right]_{i,j} - \frac{1}{\sqrt{n_T}}\mathbb{E}\left[\mathbf{Q}\mathbf{T}^{(3)}\mathbf{T}^{(3)\top}\mathbf{Q} + \mathbf{Q}\operatorname{Tr}\mathbf{T}^{(3)}\mathbf{T}^{(3)\top}\mathbf{Q}\right]_{i,j}$$

$$= \frac{n_1n_2}{\sqrt{n_T}}\mathbb{E}\left[\mathbf{Q}\right]_{i,j} - \frac{1}{\sqrt{n_T}}\mathbb{E}\left[(n_3+1)\mathbf{Q} + s\left(\mathbf{Q}\operatorname{Tr}\mathbf{Q} + \mathbf{Q}^2\right)\right]_{i,j}$$

where the last equality comes from $\mathbf{T}^{(3)}\mathbf{T}^{(3)\top}\mathbf{Q} = \boldsymbol{I}_{n_3} + s\mathbf{Q}$.

$$\mathbb{E}\left[\boldsymbol{z}\boldsymbol{m}^\top\mathbf{T}^{(3)\top}\mathbf{Q}\right]_{i,j} = \beta_T\mathbb{E}\left[\|\mathbf{M}\|_{\mathrm{F}}^2\,\boldsymbol{z}\boldsymbol{z}^\top\mathbf{Q}\right]_{i,j} + \frac{1}{\sqrt{n_T}}\sum_{k,l=1}^{n_1n_2,n_3}\mathbb{E}\left[z_im_k\mathrm{W}_{l,k}^{(3)}\mathbf{Q}_{l,j}\right]$$

$$= \beta_T\mathbb{E}\left[\|\mathbf{M}\|_{\mathrm{F}}^2\,\boldsymbol{z}\boldsymbol{z}^\top\mathbf{Q}\right]_{i,j} + \frac{1}{\sqrt{n_T}}\sum_{k,l=1}^{n_1n_2,n_3}\mathbb{E}\left[z_im_k\frac{\partial\mathbf{Q}_{l,j}}{\partial\mathrm{W}_{l,k}^{(3)}}\right]$$

$$= \beta_T\mathbb{E}\left[\|\mathbf{M}\|_{\mathrm{F}}^2\,\boldsymbol{z}\boldsymbol{z}^\top\mathbf{Q}\right]_{i,j}$$

$$\quad - \frac{1}{n_T}\sum_{k,l=1}^{n_1n_2,n_3}\mathbb{E}\left[z_im_k\left(\mathbf{Q}_{l,l}\left[\mathbf{T}^{(3)\top}\mathbf{Q}\right]_{k,j} + \mathbf{Q}_{l,j}\left[\mathbf{T}^{(3)\top}\mathbf{Q}\right]_{k,l}\right)\right]$$

$$\mathbb{E}\left[\boldsymbol{z}\boldsymbol{m}^\top\mathbf{T}^{(3)\top}\mathbf{Q}\right]_{i,j} = \beta_T\mathbb{E}\left[\|\mathbf{M}\|_{\mathrm{F}}^2\,\boldsymbol{z}\boldsymbol{z}^\top\mathbf{Q}\right]_{i,j} - \frac{1}{n_T}\mathbb{E}\left[\boldsymbol{z}\boldsymbol{m}^\top\mathbf{T}^{(3)\top}\left(\mathbf{Q}\operatorname{Tr}\mathbf{Q} + \mathbf{Q}^2\right)\right]_{i,j}.$$

$\square$

### E.2 A FIRST MATRIX EQUIVALENT

Since $\mathbf{Q}^{-1}\mathbf{Q} = \boldsymbol{I}_{n_3}$,

$$\beta_T \boldsymbol{z} \boldsymbol{m}^\top \mathbf{T}^{(3)\top}\mathbf{Q} + \frac{1}{\sqrt{n_T}}\mathbf{W}^{(3)}\mathbf{T}^{(3)\top}\mathbf{Q} - s\mathbf{Q} = \boldsymbol{I}_{n_3}$$

and, from equation 18, we have

$$\beta_T \mathbb{E}\left[\boldsymbol{z}\boldsymbol{m}^\top \mathbf{T}^{(3)\top}\mathbf{Q}\right] + \frac{n_1 n_2}{n_T}\mathbb{E}\left[\mathbf{Q}\right] - \frac{1}{n_T}\mathbb{E}\left[(n_3 + 1)\mathbf{Q} + s\left(\mathbf{Q}^2 + \mathbf{Q}\operatorname{Tr}\mathbf{Q}\right)\right] - s\mathbb{E}\left[\mathbf{Q}\right] = \boldsymbol{I}_{n_3}.$$

Let us rearrange the terms

$$s\frac{n_3}{n_T}\mathbb{E}\left[\frac{\operatorname{Tr}\mathbf{Q}}{n_3}\mathbf{Q}\right] + \left(s + \frac{n_3 - n_1 n_2}{n_T}\right)\mathbb{E}\left[\mathbf{Q}\right] + \boldsymbol{I}_{n_3} = \beta_T \mathbb{E}\left[\boldsymbol{z}\boldsymbol{m}^\top \mathbf{T}^{(3)\top}\mathbf{Q}\right] - \frac{1}{n_T}\mathbb{E}\left[\mathbf{Q} + s\mathbf{Q}^2\right]$$

in order to see that we need to following rescaling to counteract the divergence of the spectrum of $\mathbf{T}^{(3)}\mathbf{T}^{(3)\top}$,

$$\tilde{s} = \frac{n_T s - (n_3 + n_1 n_2)}{\sqrt{n_1 n_2 n_3}}, \qquad \tilde{\mathbf{Q}}(\tilde{s}) = \left(\frac{n_T \mathbf{T}^{(3)}\mathbf{T}^{(3)\top} - (n_3 + n_1 n_2)\boldsymbol{I}_{n_3}}{\sqrt{n_1 n_2 n_3}} - \tilde{s}\boldsymbol{I}_{n_3}\right)^{-1}.$$

Hence, our equation becomes,

$$\left(\frac{n_T}{\sqrt{n_1 n_2 n_3}}\tilde{s} + \frac{n_T}{n_1 n_2} + \frac{n_T}{n_3}\right)\frac{n_3}{n_T}\mathbb{E}\left[\frac{\operatorname{Tr}\tilde{\mathbf{Q}}}{n_3}\tilde{\mathbf{Q}}\right] + \left(\tilde{s} + \frac{2n_3}{\sqrt{n_1 n_2 n_3}}\right)\mathbb{E}\left[\tilde{\mathbf{Q}}\right] + \boldsymbol{I}_{n_3}$$

$$= \frac{\beta_T n_T}{\sqrt{n_1 n_2 n_3}}\mathbb{E}\left[\boldsymbol{z}\boldsymbol{m}^\top \mathbf{T}^{(3)\top}\tilde{\mathbf{Q}}\right] - \frac{1}{\sqrt{n_1 n_2 n_3}}\mathbb{E}\left[\tilde{\mathbf{Q}} + \left(\tilde{s} + \frac{n_3 + n_1 n_2}{\sqrt{n_1 n_2 n_3}}\right)\tilde{\mathbf{Q}}^2\right]. \quad (20)$$

Moreover, equation 19 becomes,

$$\mathbb{E}\left[\boldsymbol{z}\boldsymbol{m}^\top \mathbf{T}^{(3)\top}\left(\tilde{\mathbf{Q}} + \frac{1}{\sqrt{n_1 n_2 n_3}}\left(\tilde{\mathbf{Q}}\operatorname{Tr}\tilde{\mathbf{Q}} + \tilde{\mathbf{Q}}^2\right)\right)\right] = \beta_T \mathbb{E}\left[\|\mathbf{M}\|_{\mathrm{F}}^2 \boldsymbol{z}\boldsymbol{z}^\top \tilde{\mathbf{Q}}\right]. \quad (21)$$

Therefore, denoting $\tilde{m}(\tilde{s}) = \lim \frac{\operatorname{Tr}\tilde{\mathbf{Q}}(\tilde{s})}{n_3}$ the Stieltjes transform of the limiting spectral distribution of $\frac{n_T}{\sqrt{n_1 n_2 n_3}}\mathbf{T}^{(3)}\mathbf{T}^{(3)\top} - \frac{n_3 + n_1 n_2}{\sqrt{n_1 n_2 n_3}}\boldsymbol{I}_{n_3}$, the combination of equation 20 and equation 21 yield the following matrix equivalent,

$$\tilde{m}(\tilde{s})\tilde{\mathbf{Q}} + \tilde{s}\tilde{\mathbf{Q}} + \boldsymbol{I}_{n_3} \longleftrightarrow \frac{\beta_T^2 n_T}{\sqrt{n_1 n_2 n_3}}\|\mathbf{M}\|_{\mathrm{F}}^2 \boldsymbol{z}\boldsymbol{z}^\top \tilde{\mathbf{Q}}. \quad (22)$$

At this point, we are still carrying the random term $\|\mathbf{M}\|_{\mathrm{F}}^2$ in the RHS. This quantity is expected to concentrate rapidly around its mean $\beta_M^2 + \frac{n_1 n_2}{n_M}$. However, because $\mathbf{M}$ and $\tilde{\mathbf{Q}}$ are not independent, the RHS must be studied carefully.

### E.3 ANALYSIS OF THE RHS

**Lemma 3.**

$$\frac{\partial \mathbf{Q}_{a,b}}{\partial \mathbf{Z}_{c,d}} = -\frac{\beta_T}{\sqrt{n_M}}\left([\mathbf{Q}\boldsymbol{z}]_a \left[\mathbf{T}^{(3)\top}\mathbf{Q}\right]_{[c,d],b} + [\mathbf{Q}\boldsymbol{z}]_b \left[\mathbf{T}^{(3)\top}\mathbf{Q}\right]_{[c,d],a}\right).$$

*Proof.* Since $\partial\mathbf{Q} = -\mathbf{Q}\partial\left(\mathbf{T}^{(3)}\mathbf{T}^{(3)\top}\right)\mathbf{Q}$,

$$\frac{\partial \mathbf{Q}_{a,b}}{\partial \mathbf{Z}_{c,d}} = -\sum_{e,f,g,h=1}^{n_3, n_1, n_2, n_3} \mathbf{Q}_{a,e}\left(\frac{\partial \mathcal{T}_{f,g,e}}{\partial \mathbf{Z}_{c,d}}\mathcal{T}_{f,g,h} + \mathcal{T}_{f,g,e}\frac{\partial \mathcal{T}_{f,g,h}}{\partial \mathbf{Z}_{c,d}}\right)\mathbf{Q}_{h,b}$$

$$= -\frac{\beta_T}{\sqrt{n_M}}\sum_{e,h=1}^{n_3}\mathbf{Q}_{a,e}\left(z_e \mathcal{T}_{c,d,h} + \mathcal{T}_{c,d,e}z_h\right)\mathbf{Q}_{h,b}$$

$$\frac{\partial \mathbf{Q}_{a,b}}{\partial \mathbf{Z}_{c,d}} = -\frac{\beta_T}{\sqrt{n_M}}\left([\mathbf{Q}\boldsymbol{z}]_a \left[\mathbf{T}^{(3)\top}\mathbf{Q}\right]_{[c,d],b} + [\mathbf{Q}\boldsymbol{z}]_b \left[\mathbf{T}^{(3)\top}\mathbf{Q}\right]_{[c,d],a}\right).$$

$\square$

$$\mathbb{E}\left[\|\mathbf{M}\|_{\mathrm{F}}^2\,\mathbf{Q}\right] = \sum_{u,v=1}^{n_1,n_2}\mathbb{E}\left[\left(\beta_M x_u y_v + \frac{1}{\sqrt{n_M}}\mathbf{Z}_{u,v}\right)^2\mathbf{Q}\right]$$

$$= \beta_M^2\,\mathbb{E}\left[\mathbf{Q}\right] + \frac{1}{\sqrt{n_M}}\sum_{u,v=1}^{n_1,n_2}\mathbb{E}\left[\left(\frac{\mathbf{Z}_{u,v}}{\sqrt{n_M}} + 2\beta_M x_u y_v\right)\mathbf{Z}_{u,v}\mathbf{Q}\right]$$

$$\mathbb{E}\left[\|\mathbf{M}\|_{\mathrm{F}}^2\,\mathbf{Q}\right] = \left(\beta_M^2 + \frac{n_1 n_2}{n_M}\right)\mathbb{E}\left[\mathbf{Q}\right] + \frac{1}{\sqrt{n_M}}\sum_{u,v=1}^{n_1,n_2}\mathbb{E}\left[\left(\frac{\mathbf{Z}_{u,v}}{\sqrt{n_M}} + 2\beta_M x_u y_v\right)\frac{\partial\mathbf{Q}}{\partial\mathbf{Z}_{u,v}}\right]$$

where we have used Stein's lemma (Lemma 1) to derive the last equality. Hence, using Lemma 3, it becomes

$$\mathbb{E}\left[\left(\|\mathbf{M}\|_{\mathrm{F}}^2 - \left(\beta_M^2 + \frac{n_1 n_2}{n_M}\right)\right)\mathbf{Q}\right]_{i,j}$$

$$= -\frac{\beta_T}{n_M}\sum_{u,v=1}^{n_1,n_2}\mathbb{E}\left[\left(\frac{\mathbf{Z}_{u,v}}{\sqrt{n_M}} + 2\beta_M x_u y_v\right)\left([\mathbf{Q}z]_i\left[\mathbf{T}^{(3)\top}\mathbf{Q}\right]_{[u,v],j} + [\mathbf{Q}z]_j\left[\mathbf{T}^{(3)\top}\mathbf{Q}\right]_{[u,v],i}\right)\right]$$

$$= -\frac{\beta_T}{n_M}\sum_{u,v=1}^{n_1,n_2}\mathbb{E}\left[\left(2m_{[u,v]} - \frac{\mathbf{Z}_{u,v}}{\sqrt{n_M}}\right)\left([\mathbf{Q}z]_i\left[\mathbf{T}^{(3)\top}\mathbf{Q}\right]_{[u,v],j} + [\mathbf{Q}z]_j\left[\mathbf{T}^{(3)\top}\mathbf{Q}\right]_{[u,v],i}\right)\right]$$

$$= -2\frac{\beta_T}{n_M}\mathbb{E}\left[\mathbf{Q}zm^\top\mathbf{T}^{(3)\top}\mathbf{Q} + \left(\mathbf{Q}zm^\top\mathbf{T}^{(3)\top}\mathbf{Q}\right)^\top\right]_{i,j}$$

$$+ \frac{\beta_T}{n_M\sqrt{n_M}}\sum_{u,v=1}^{n_1,n_2}\mathbb{E}\left[\mathbf{Z}_{u,v}\left([\mathbf{Q}z]_i\left[\mathbf{T}^{(3)\top}\mathbf{Q}\right]_{[u,v],j} + [\mathbf{Q}z]_j\left[\mathbf{T}^{(3)\top}\mathbf{Q}\right]_{[u,v],i}\right)\right]$$

and, since $\beta_T zm^\top = \mathbf{T}^{(3)} - \frac{1}{\sqrt{n_T}}\mathbf{W}^{(3)}$ and $\mathbf{T}^{(3)}\mathbf{T}^{(3)\top}\mathbf{Q} = \boldsymbol{I}_{n_3} + s\mathbf{Q}$, we have,

$$\beta_T zm^\top\mathbf{T}^{(3)\top}\mathbf{Q} = \boldsymbol{I}_{n_3} + s\mathbf{Q} - \frac{1}{\sqrt{n_T}}\mathbf{W}^{(3)}\mathbf{T}^{(3)\top}\mathbf{Q}.$$

Thus, our expression turns into

$$\mathbb{E}\left[\left(\|\mathbf{M}\|_{\mathrm{F}}^2 - \left(\beta_M^2 + \frac{n_1 n_2}{n_M}\right)\right)\mathbf{Q}\right]_{i,j}$$

$$= -\frac{4}{n_M}\mathbb{E}\left[\mathbf{Q} + s\mathbf{Q}^2\right]_{i,j} + \frac{2}{n_M\sqrt{n_T}}\mathbb{E}\left[\mathbf{Q}\left(\mathbf{W}^{(3)}\mathbf{T}^{(3)\top} + \mathbf{T}^{(3)}\mathbf{W}^{(3)\top}\right)\mathbf{Q}\right]_{i,j}$$

$$+ \frac{\beta_T}{n_M\sqrt{n_M}}\sum_{u,v=1}^{n_1,n_2}\mathbb{E}\left[\mathbf{Z}_{u,v}\left([\mathbf{Q}z]_i\left[\mathbf{T}^{(3)\top}\mathbf{Q}\right]_{[u,v],j} + [\mathbf{Q}z]_j\left[\mathbf{T}^{(3)\top}\mathbf{Q}\right]_{[u,v],i}\right)\right]$$

$$= -\frac{4}{n_M}\mathbb{E}\left[\mathbf{Q} + s\mathbf{Q}^2\right]_{i,j} + \frac{2}{n_M\sqrt{n_T}}\left[\boldsymbol{A}_1 + \boldsymbol{A}_1^\top\right]_{i,j} + \frac{\beta_T}{n_M\sqrt{n_M}}\left[\boldsymbol{A}_2 + \boldsymbol{A}_2^\top\right]_{i,j}$$

with

$$\boldsymbol{A}_1 = \mathbb{E}\left[\mathbf{Q}\mathbf{W}^{(3)}\mathbf{T}^{(3)\top}\mathbf{Q}\right], \qquad [\boldsymbol{A}_2]_{i,j} = \sum_{u,v=1}^{n_1,n_2}\mathbb{E}\left[\mathbf{Z}_{u,v}[\mathbf{Q}z]_i\left[\mathbf{T}^{(3)\top}\mathbf{Q}\right]_{[u,v],j}\right].$$

Let us develop $\boldsymbol{A}_1$ with Stein's lemma (Lemma 1) on $\mathbf{W}^{(3)}$.

$$
\begin{aligned}
[\boldsymbol{A}_1]_{i,j} &= \sum_{a,b,c=1}^{n_3,n_1 n_2,n_3} \mathbb{E}\left[ \frac{\partial \mathbf{Q}_{i,a}}{\partial \mathbf{W}_{a,b}^{(3)}} \mathrm{T}_{c,b}^{(3)} \mathbf{Q}_{c,j} + \mathbf{Q}_{i,a} \frac{\partial \mathrm{T}_{c,b}^{(3)}}{\partial \mathbf{W}_{a,b}^{(3)}} \mathbf{Q}_{c,j} + \mathbf{Q}_{i,a} \mathrm{T}_{c,b}^{(3)} \frac{\partial \mathbf{Q}_{c,j}}{\partial \mathbf{W}_{a,b}^{(3)}} \right] \\
&= \frac{n_1 n_2}{\sqrt{n_T}} \mathbb{E}\left[\mathbf{Q}^2\right]_{i,j} \\
&\quad - \frac{1}{\sqrt{n_T}} \sum_{a,b,c=1}^{n_3,n_1 n_2,n_3} \mathbb{E}\left[ \left( \mathbf{Q}_{i,a} \left[\mathbf{T}^{(3)\top}\mathbf{Q}\right]_{b,a} + \mathbf{Q}_{a,a} \left[\mathbf{T}^{(3)\top}\mathbf{Q}\right]_{b,i} \right) \mathrm{T}_{c,b}^{(3)} \mathbf{Q}_{c,j} \right] \\
&\quad - \frac{1}{\sqrt{n_T}} \sum_{a,b,c=1}^{n_3,n_1 n_2,n_3} \mathbb{E}\left[ \mathbf{Q}_{i,a} \mathrm{T}_{c,b}^{(3)} \left( \mathbf{Q}_{c,a} \left[\mathbf{T}^{(3)\top}\mathbf{Q}\right]_{b,j} + \mathbf{Q}_{a,j} \left[\mathbf{T}^{(3)\top}\mathbf{Q}\right]_{b,c} \right) \right] \\
&= \frac{n_1 n_2}{\sqrt{n_T}} \mathbb{E}\left[\mathbf{Q}^2\right]_{i,j} - \frac{1}{\sqrt{n_T}} \mathbb{E}\left[ \left(2\mathbf{Q}^2 + \mathbf{Q}\operatorname{Tr}\mathbf{Q}\right) \mathbf{T}^{(3)}\mathbf{T}^{(3)\top}\mathbf{Q} + \mathbf{Q}^2 \operatorname{Tr}\mathbf{T}^{(3)}\mathbf{T}^{(3)\top}\mathbf{Q} \right]_{i,j} \\
[\boldsymbol{A}_1]_{i,j} &= \frac{n_1 n_2}{\sqrt{n_T}} \mathbb{E}\left[\mathbf{Q}^2\right]_{i,j} - \frac{1}{\sqrt{n_T}} \mathbb{E}\left[ (n_3+2)\mathbf{Q}^2 + \mathbf{Q}\operatorname{Tr}\mathbf{Q} + 2s\mathbf{Q}\left(\mathbf{Q}^2 + \mathbf{Q}\operatorname{Tr}\mathbf{Q}\right) \right]_{i,j}
\end{aligned}
$$

since $\mathbf{T}^{(3)}\mathbf{T}^{(3)\top}\mathbf{Q} = \boldsymbol{I}_{n_3} + s\mathbf{Q}$.

Next, we develop $\boldsymbol{A}_2$ with Stein's lemma (Lemma 1) on $\mathbf{Z}$.

$$
\begin{aligned}
[\boldsymbol{A}_2]_{i,j} &= \sum_{u,v=1}^{n_1,n_2} \sum_{a,b=1}^{n_3,n_3} \mathbb{E}\left[ \frac{\partial \mathbf{Q}_{i,a}}{\partial \mathbf{Z}_{u,v}} z_a \mathrm{T}_{b,[u,v]}^{(3)} \mathbf{Q}_{b,j} + \mathbf{Q}_{i,a} z_a \frac{\partial \mathrm{T}_{b,[u,v]}^{(3)}}{\partial \mathbf{Z}_{u,v}} \mathbf{Q}_{b,j} + \mathbf{Q}_{i,a} z_a \mathrm{T}_{b,[u,v]}^{(3)} \frac{\partial \mathbf{Q}_{b,j}}{\partial \mathbf{Z}_{u,v}} \right] \\
&= \frac{\beta_T n_1 n_2}{\sqrt{n_M}} \mathbb{E}\left[\mathbf{Q}\boldsymbol{z}\boldsymbol{z}^\top\mathbf{Q}\right]_{i,j} \\
&\quad - \frac{\beta_T}{\sqrt{n_M}} \sum_{u,v=1}^{n_1,n_2} \sum_{a,b=1}^{n_3,n_3} \mathbb{E}\left[ [\mathbf{Q}\boldsymbol{z}]_i \left[\mathbf{T}^{(3)\top}\mathbf{Q}\right]_{[u,v],a} z_a \mathrm{T}_{b,[u,v]}^{(3)} \mathbf{Q}_{b,j} \right] \\
&\quad - \frac{\beta_T}{\sqrt{n_M}} \sum_{u,v=1}^{n_1,n_2} \sum_{a,b=1}^{n_3,n_3} \mathbb{E}\left[ [\mathbf{Q}\boldsymbol{z}]_a \left[\mathbf{T}^{(3)\top}\mathbf{Q}\right]_{[u,v],i} z_a \mathrm{T}_{b,[u,v]}^{(3)} \mathbf{Q}_{b,j} \right] \\
&\quad - \frac{\beta_T}{\sqrt{n_M}} \sum_{u,v=1}^{n_1,n_2} \sum_{a,b=1}^{n_3,n_3} \mathbb{E}\left[ \mathbf{Q}_{i,a} z_a \mathrm{T}_{b,[u,v]}^{(3)} [\mathbf{Q}\boldsymbol{z}]_b \left[\mathbf{T}^{(3)\top}\mathbf{Q}\right]_{[u,v],j} \right] \\
&\quad - \frac{\beta_T}{\sqrt{n_M}} \sum_{u,v=1}^{n_1,n_2} \sum_{a,b=1}^{n_3,n_3} \mathbb{E}\left[ \mathbf{Q}_{i,a} z_a \mathrm{T}_{b,[u,v]}^{(3)} [\mathbf{Q}\boldsymbol{z}]_j \left[\mathbf{T}^{(3)\top}\mathbf{Q}\right]_{[u,v],b} \right] \\
&= \frac{\beta_T n_1 n_2}{\sqrt{n_M}} \mathbb{E}\left[\mathbf{Q}\boldsymbol{z}\boldsymbol{z}^\top\mathbf{Q}\right]_{i,j} - \frac{\beta_T}{\sqrt{n_M}} \mathbb{E}\left[ \left(\mathbf{Q}\boldsymbol{z}\boldsymbol{z}^\top\mathbf{Q} + [\boldsymbol{z}^\top\mathbf{Q}\boldsymbol{z}]\mathbf{Q}\right) \mathbf{T}^{(3)}\mathbf{T}^{(3)\top}\mathbf{Q} \right]_{i,j} \\
&\quad - \frac{\beta_T}{\sqrt{n_M}} \mathbb{E}\left[ \mathbf{Q}\boldsymbol{z}\boldsymbol{z}^\top \left(\mathbf{Q}\mathbf{T}^{(3)}\mathbf{T}^{(3)\top}\mathbf{Q} + \mathbf{Q}\operatorname{Tr}\mathbf{T}^{(3)}\mathbf{T}^{(3)\top}\mathbf{Q}\right) \right]_{i,j} \\
[\boldsymbol{A}_2]_{i,j} &= \frac{\beta_T n_1 n_2}{\sqrt{n_M}} \mathbb{E}\left[\mathbf{Q}\boldsymbol{z}\boldsymbol{z}^\top\mathbf{Q}\right]_{i,j} - \frac{\beta_T}{\sqrt{n_M}} \mathbb{E}\left[ (n_3+2)\mathbf{Q}\boldsymbol{z}\boldsymbol{z}^\top\mathbf{Q} + [\boldsymbol{z}^\top\mathbf{Q}\boldsymbol{z}]\mathbf{Q} \right]_{i,j} \\
&\quad - \frac{\beta_T}{\sqrt{n_M}} s\mathbb{E}\left[ \mathbf{Q}\boldsymbol{z}\boldsymbol{z}^\top \left(\mathbf{Q}\operatorname{Tr}\mathbf{Q} + 2\mathbf{Q}^2\right) + [\boldsymbol{z}^\top\mathbf{Q}\boldsymbol{z}]\mathbf{Q}^2 \right]_{i,j}
\end{aligned}
$$

since, again, $\mathbf{T}^{(3)}\mathbf{T}^{(3)\top}\mathbf{Q} = \boldsymbol{I}_{n_3} + s\mathbf{Q}$.

Eventually, we have,

$$\mathbb{E}\left[\|\mathbf{M}\|_{\mathrm{F}}^2 \mathbf{Q}\right] = \left(\beta_M^2 + \frac{n_1 n_2}{n_M}\right)\mathbb{E}\left[\mathbf{Q}\right] - \frac{4}{n_M}\mathbb{E}\left[\mathbf{Q} + s\mathbf{Q}^2\right]$$
$$+ \frac{4n_1 n_2}{n_M n_T}\mathbb{E}\left[\mathbf{Q}^2\right] - \frac{4}{n_M n_T}\mathbb{E}\left[(n_3 + 2)\mathbf{Q}^2 + \mathbf{Q}\operatorname{Tr}\mathbf{Q} + 2s\mathbf{Q}\left(\mathbf{Q}^2 + \mathbf{Q}\operatorname{Tr}\mathbf{Q}\right)\right]$$
$$+ \frac{2\beta_T^2 n_1 n_2}{n_M^2}\mathbb{E}\left[\mathbf{Q}zz^\top\mathbf{Q}\right] - \frac{2\beta_T^2}{n_M^2}\mathbb{E}\left[(n_3 + 2)\mathbf{Q}zz^\top\mathbf{Q} + \left[z^\top\mathbf{Q}z\right]\mathbf{Q}\right]$$
$$- \frac{2\beta_T^2}{n_M^2}s\mathbb{E}\left[\mathbf{Q}zz^\top\mathbf{Q}\operatorname{Tr}\mathbf{Q} + \left[z^\top\mathbf{Q}z\right]\mathbf{Q}^2\right] - \frac{2\beta_T^2}{n_M^2}s\mathbb{E}\left[\mathbf{Q}zz^\top\mathbf{Q}^2 + \mathbf{Q}^2 zz^\top\mathbf{Q}\right]$$

$$\mathbb{E}\left[\|\mathbf{M}\|_{\mathrm{F}}^2 \mathbf{Q}\right] = \left(\frac{n_1 n_2}{n_M} + \beta_M^2 - \frac{4}{n_M}\right)\mathbb{E}\left[\mathbf{Q}\right] - \frac{4}{n_M}\left(s + \frac{n_3 - n_1 n_2}{n_T} + \frac{2}{n_T}\right)\mathbb{E}\left[\mathbf{Q}^2\right]$$
$$- \frac{4n_3}{n_M n_T}\mathbb{E}\left[\mathbf{Q}\frac{\operatorname{Tr}\mathbf{Q}}{n_3}\right] - \frac{8}{n_M n_T}s\mathbb{E}\left[\mathbf{Q}^3\right] - \frac{8n_3}{n_M n_T}s\mathbb{E}\left[\mathbf{Q}^2\frac{\operatorname{Tr}\mathbf{Q}}{n_3}\right]$$
$$- \frac{2\beta_T^2}{n_M^2}\mathbb{E}\left[(n_3 - n_1 n_2 + 2)\mathbf{Q}zz^\top\mathbf{Q} + \left[z^\top\mathbf{Q}z\right]\mathbf{Q}\right]$$
$$- \frac{2\beta_T^2}{n_M^2}s\mathbb{E}\left[n_3\mathbf{Q}zz^\top\mathbf{Q}\frac{\operatorname{Tr}\mathbf{Q}}{n_3} + \left[z^\top\mathbf{Q}z\right]\mathbf{Q}^2 + \mathbf{Q}zz^\top\mathbf{Q}^2 + \mathbf{Q}^2 zz^\top\mathbf{Q}\right].$$

After rescaling, this becomes,

$$\frac{n_T}{\sqrt{n_1 n_2 n_3}}\mathbb{E}\left[\|\mathbf{M}\|_{\mathrm{F}}^2 \tilde{\mathbf{Q}}\right]$$
$$= \left(\frac{n_1 n_2}{n_M} + \beta_M^2 - \frac{4}{n_M}\right)\frac{n_T}{\sqrt{n_1 n_2 n_3}}\mathbb{E}\left[\tilde{\mathbf{Q}}\right] - \frac{4}{n_M}\left(\tilde{s} + \frac{2(n_3 + 1)}{\sqrt{n_1 n_2 n_3}}\right)\frac{n_T}{\sqrt{n_1 n_2 n_3}}\mathbb{E}\left[\tilde{\mathbf{Q}}^2\right]$$
$$- \frac{4n_T}{n_M n_1 n_2}\mathbb{E}\left[\tilde{\mathbf{Q}}\frac{\operatorname{Tr}\tilde{\mathbf{Q}}}{n_3}\right] - \frac{8n_T}{n_M n_1 n_2}\left(\tilde{s} + \frac{n_3 + n_1 n_2}{\sqrt{n_1 n_2 n_3}}\right)\mathbb{E}\left[\frac{1}{n_3}\tilde{\mathbf{Q}}^3 + \tilde{\mathbf{Q}}^2\frac{\operatorname{Tr}\tilde{\mathbf{Q}}}{n_3}\right]$$
$$- \frac{2\beta_T^2 n_T^2}{n_M^2 n_1 n_2 n_3}\mathbb{E}\left[(n_3 - n_1 n_2 + 2)\tilde{\mathbf{Q}}zz^\top\tilde{\mathbf{Q}} + \left[z^\top\tilde{\mathbf{Q}}z\right]\tilde{\mathbf{Q}}\right]$$
$$- \frac{2\beta_T^2 n_T^2}{n_M^2 n_1 n_2 n_3}\left(\tilde{s} + \frac{n_3 + n_1 n_2}{\sqrt{n_1 n_2 n_3}}\right)\mathbb{E}\left[n_3\tilde{\mathbf{Q}}zz^\top\tilde{\mathbf{Q}}\frac{\operatorname{Tr}\tilde{\mathbf{Q}}}{n_3}\right]$$
$$- \frac{2\beta_T^2 n_T^2}{n_M^2 n_1 n_2 n_3}\left(\tilde{s} + \frac{n_3 + n_1 n_2}{\sqrt{n_1 n_2 n_3}}\right)\mathbb{E}\left[\left[z^\top\tilde{\mathbf{Q}}z\right]\tilde{\mathbf{Q}}^2 + \tilde{\mathbf{Q}}zz^\top\tilde{\mathbf{Q}}^2 + \tilde{\mathbf{Q}}^2 zz^\top\tilde{\mathbf{Q}}\right]$$
$$= \Theta\left(\sqrt{n_T} + \frac{\beta_M^2}{\sqrt{n_T}}\right)\mathbb{E}\left[\tilde{\mathbf{Q}}\right] - \Theta\left(\frac{1}{n_T\sqrt{n_T}}\right)\mathbb{E}\left[\tilde{\mathbf{Q}}^2\right]$$
$$- \Theta\left(\frac{1}{n_T^2}\right)\mathbb{E}\left[\tilde{\mathbf{Q}}\frac{\operatorname{Tr}\tilde{\mathbf{Q}}}{n_3}\right] - \Theta\left(\frac{1}{n_T^2\sqrt{n_T}}\right)\mathbb{E}\left[\tilde{\mathbf{Q}}^3\right] - \Theta\left(\frac{1}{n_T\sqrt{n_T}}\right)\mathbb{E}\left[\tilde{\mathbf{Q}}^2\frac{\operatorname{Tr}\tilde{\mathbf{Q}}}{n_3}\right]$$
$$- \Theta\left(\frac{\beta_T^2}{n_T}\right)\mathbb{E}\left[\tilde{\mathbf{Q}}zz^\top\tilde{\mathbf{Q}}\right] - \Theta\left(\frac{\beta_T^2}{n_T^3}\right)\mathbb{E}\left[\left[z^\top\tilde{\mathbf{Q}}z\right]\tilde{\mathbf{Q}}\right]$$
$$- \Theta\left(\frac{\beta_T^2}{n_T\sqrt{n_T}}\right)\mathbb{E}\left[\tilde{\mathbf{Q}}zz^\top\tilde{\mathbf{Q}}\frac{\operatorname{Tr}\tilde{\mathbf{Q}}}{n_3}\right]$$
$$- \Theta\left(\frac{\beta_T^2}{n_T^2\sqrt{n_T}}\right)\mathbb{E}\left[\left[z^\top\tilde{\mathbf{Q}}z\right]\tilde{\mathbf{Q}}^2 + \tilde{\mathbf{Q}}zz^\top\tilde{\mathbf{Q}}^2 + \tilde{\mathbf{Q}}^2 zz^\top\tilde{\mathbf{Q}}\right].$$

Hence, as long as $\beta_T = o(n_T^{3/4})$, we have

$$\frac{\beta_T^2 n_T}{\sqrt{n_1 n_2 n_3}}\|\mathbf{M}\|_{\mathrm{F}}^2 \tilde{\mathbf{Q}} \longleftrightarrow \beta_T^2\left(\frac{n_T}{n_M}\sqrt{\frac{n_1 n_2}{n_3}} + \frac{\beta_M^2 n_T}{\sqrt{n_1 n_2 n_3}}\right)\tilde{\mathbf{Q}}.$$

### E.4 DETERMINISTIC EQUIVALENT

Coming back to equation 22, we are now allowed to write,

$$\tilde{m}(\tilde{s})\tilde{\mathbf{Q}} + \tilde{s}\tilde{\mathbf{Q}} + \mathbf{I}_{n_3} \longleftrightarrow \beta_T^2 \left( \frac{n_T}{n_M} \sqrt{\frac{n_1 n_2}{n_3}} + \frac{\beta_M^2 n_T}{\sqrt{n_1 n_2 n_3}} \right) \mathbf{z}\mathbf{z}^\top \tilde{\mathbf{Q}}.$$

Hence, in order to prevent the RHS from either vanishing or exploding, we see that $\beta_T$ must be $\Theta(n_T^{-1/4})$. In this case, Let $\varrho = \lim \beta_T^2 \left( \frac{n_T}{n_M} \sqrt{\frac{n_1 n_2}{n_3}} + \frac{\beta_M^2 n_T}{\sqrt{n_1 n_2 n_3}} \right)$.

Then, we can define $\bar{\mathbf{Q}}$, the deterministic equivalent of $\tilde{\mathbf{Q}}$ such that,

$$\tilde{m}(\tilde{s})\bar{\mathbf{Q}} + \tilde{s}\bar{\mathbf{Q}} + \mathbf{I}_{n_3} = \varrho \mathbf{z}\mathbf{z}^\top \bar{\mathbf{Q}}. \tag{23}$$

*Remark.* In fact, with the assumption $\beta_M = \Theta(1)$, $\beta_T^2 \left( \frac{n_T}{n_M} \sqrt{\frac{n_1 n_2}{n_3}} + \frac{\beta_M^2 n_T}{\sqrt{n_1 n_2 n_3}} \right) \sim \beta_T^2 \frac{n_T}{n_M} \sqrt{\frac{n_1 n_2}{n_3}}$ so we could have kept only this dominant term. However, for finite horizon considerations (as it is the case in practice), adding the $\Theta\left( \frac{\beta_M^2}{\sqrt{n_T}} \right)$ term leads to slightly more precise predictions. Indeed, with the dominant term only, we consider a "worst-case scenario" $\beta_M = 0$.

### E.5 LIMITING SPECTRAL DISTRIBUTION

According to the previous deterministic equivalent, the Stieltjes transform of the limiting spectral distribution of $\frac{n_T}{\sqrt{n_1 n_2 n_3}} \mathbf{T}^{(3)} \mathbf{T}^{(3)\top} - \frac{n_3 + n_1 n_2}{\sqrt{n_1 n_2 n_3}} \mathbf{I}_{n_3}$ is solution to

$$\tilde{m}^2(\tilde{s}) + \tilde{s}\tilde{m}(\tilde{s}) + 1 = 0, \tag{24}$$

i.e., it is the standard semi-circle law.

## F PROOF OF THEOREM 7

### F.1 ISOLATED EIGENVALUE

The asymptotic position $\tilde{\xi}$ of the isolated eigenvalue (when it exists) is a singular point of $\bar{\mathbf{Q}}$. From equation 23, it is therefore solution to

$$\tilde{m}(\tilde{\xi}) + \tilde{\xi} = \varrho.$$

Following equation 24, $\tilde{\xi} = -\tilde{m}(\tilde{\xi}) - \frac{1}{\tilde{m}(\tilde{\xi})}$, thus, $\tilde{m}(\tilde{\xi}) = -\frac{1}{\varrho}$. Injecting this in equation 24 yields

$$\tilde{\xi} = \varrho + \frac{1}{\varrho}. \tag{25}$$

### F.2 EIGENVECTOR ALIGNMENT

Following Cauchy's integral formula,

$$\left| \mathbf{z}^\top \hat{\mathbf{z}} \right|^2 = -\frac{1}{2i\pi} \oint_{\tilde{\gamma}} \mathbf{z}^\top \tilde{\mathbf{Q}}(\tilde{s}) \mathbf{z} \, \mathrm{d}\tilde{s}$$

where $\tilde{\gamma}$ is a positively-oriented complex contour circling around the isolated eigenvalue only. Hence, using the previously found deterministic equivalent $\bar{\mathbf{Q}}$, we can compute the asymptotic value $\zeta$ of $\left| \mathbf{z}^\top \hat{\mathbf{z}} \right|^2$,

$$\zeta = -\frac{1}{2i\pi} \oint_{\tilde{\gamma}} \mathbf{z}^\top \bar{\mathbf{Q}}(\tilde{s}) \mathbf{z} \, \mathrm{d}\tilde{s}.$$

This reduces to residue calculus,

$$\zeta = \lim_{\tilde{s} \to \tilde{\xi}} \left( \tilde{s} - \tilde{\xi} \right) [\tilde{m}(\tilde{s}) + \tilde{s} - \varrho]^{-1} = \frac{1}{\frac{\mathrm{d}}{\mathrm{d}\tilde{s}} [\tilde{m}(\tilde{s}) + \tilde{s} - \varrho]_{\tilde{s}=\tilde{\xi}}} = \frac{1}{\tilde{m}'(\tilde{\xi}) + 1}.$$

Differentiating equation 24, we have,

$$2\tilde{m}'(\tilde{\xi})m(\tilde{\xi}) + \tilde{m}(\tilde{\xi}) + \tilde{\xi}\tilde{m}'(\tilde{\xi}) = 0 \iff \tilde{m}'(\tilde{\xi}) = -\frac{\tilde{m}(\tilde{\xi})}{2m(\tilde{\xi}) + \tilde{\xi}}$$
$$\iff \tilde{m}'(\tilde{\xi}) = \frac{1}{\varrho\tilde{\xi} - 2}$$
$$\iff \tilde{m}'(\tilde{\xi}) = \frac{1}{\varrho^2 - 1}.$$

Hence, we conclude,

$$\zeta = 1 - \frac{1}{\varrho^2}. \tag{26}$$

