# OpenReview forum: "Performance Gaps in Multi-view Clustering under the Nested Matrix-Tensor Model"
_ICLR.cc/2024/Conference — ICLR 2024 poster_

### Official Review · Reviewer_sHnH · 2023-11-01

**Soundness:** 3 good
**Presentation:** 3 good
**Contribution:** 3 good
**Rating:** 6
**Confidence:** 1

**Summary:**

The authors quantify the gap between the best rank-one approximation of a tensor and that of its unfolding matrix.

**Strengths:**

The theoretical analysis of the paper is solid and the results is interesting.

**Weaknesses:**

The assumption that both of Z and W follow the Gaussian distribution with zero mean and unit variance, is too strong. As $X=(\boldsymbol{\mu}\boldsymbol{h}^T)\otimes\boldsymbol{\mu} + \boldsymbol{Z}\otimes\boldsymbol{\mu}+\boldsymbol{W}$, thus the variance of Z and W affect the performance of the rank-one approximation in a different manner. But the authors do not discuss the part and simply assume both Z and W follow the normal distribution.

**Questions:**

1, It is important to introduce and define all symbols used in the paper when they are first mentioned. So it is better to define symbol $\otimes$ in Nested Matrix-Tensor Model.

---

> ### Author Response · Authors · 2023-11-20
>
> Thank you for your relevant feedback, which has helped us improve the clarity of the paper.
>
> **Weaknesses**
> The assumption that $\mathbf{Z}$ and $\mathbf{W}$ are Gaussian is, in fact, not restrictive in our case, but greatly simplifies the presentation. In order to extend our results to non-Gaussian noise, see the work of Anna Lytova and Leonid Pastur, "Central limit theorem for linear eigenvalue statistics of random matrices with independent entries". Their Corollary 3.1 can be viewed as a generalized Stein's lemma (Lemma 1 in our paper) for non-Gaussian distributions. All Gaussian expectations of our proofs can then be expressed as their non-Gaussian form plus a residual term controlled by the moments of the distribution. A reference to this "interpolation trick" has been added to the presentation of the model at the beginning of Section 3.
>
> **Questions**
> We have added a clarification on the symbol $\otimes$ after its first use in the introduction.

---

### Official Review · Reviewer_hX5q · 2023-11-04

**Soundness:** 3 good
**Presentation:** 3 good
**Contribution:** 2 fair
**Rating:** 5
**Confidence:** 2

**Summary:**

This paper focuses on theoretical analysis for Multiview clustering. Specifically, the authors quantify the performance gap between the tensor-based approach and the unfolding approaches.

**Strengths:**

1.	The motivation of this paper is clear.
2.	Providing theoretical analysis for tensor-based Multiview clustering is important.
3.	The proofs seem strict.

**Weaknesses:**

1.	There are few tensor-based Multiview clustering methods mentioned, thus, it is difficult to understand which kinds of tensor-based methods are suitable for these theoretical results.

2.	The experimental results are not sufficient. For example, in practice, there are many tensor-based and matrix or vector-based Multiview clustering methods, so what about the performance gaps between them and is the theoretical results applicable for them?

3.	The potential inspiration for researchers of the work is not clear.

**Questions:**

see weakness

---

> ### Author Response · Authors · 2023-11-20
>
> Thank you for your profitable feedback. We hope that our answers will adequately address your comments.
> 1. As they are, our theoretical results specifically concern the estimation of the clusters in our nested model via either a tensor-based or a matrix-based spectral approach, that is, either best rank-1 tensor or best rank-1 matrix approximation. As stated in the Introduction, the goal is to rigorously study performance gaps between tensor-based approaches and matrix-based ones in a setting where the sought signal has tensor structure, and thus the latter method can be seen as a relaxation. This study is one example of this general idea, but more work is needed to rigorously extend its conclusions beyond the considered model.
> 2. Please note that our main goal is to give rigorous theoretical results quantifying the performance gap between tensor-based and matrix-based approaches in a specific setting, with a simplified model. Hence, the numerical results are only meant to illustrate our theoretical findings, showing their implications in practice. However, our paper does not purport to explain the performance gap between any tensor-based and any matrix-based multi-view clustering methods. Other comparisons need further study, and are out of the scope of this paper. We have outlined this point in our introduction.
> 3. The main message the paper is meant to convey is that, in the case of the nested matrix-tensor model, which serves a simplified model of multi-view clustering, we can rigorously prove that there is a significant performance gap between tensor-based and matrix-based approaches, confirming intuition and previous empirical evidence. We hope this will inspire similar works in this direction, which precisely elucidate the gain brought by taking into account the tensor structure of a given model's signal.

---

### Official Review · Reviewer_9jwW · 2023-11-04

**Soundness:** 3 good
**Presentation:** 3 good
**Contribution:** 2 fair
**Rating:** 6
**Confidence:** 2

**Summary:**

This paper studies the problem of estimating a planted signal that is hidden in a nested matrix-tensor model, which generalizes the classical spiked rank-one tensor model. The paper compares the performance of two approaches: tensor-based and matrix-based. The tensor-based approach exploits the tensor structure of the data, while the matrix-based approach uses the unfolding of the tensor. The paper derives the exact algorithmic threshold of the matrix-based approach, and shows that it undergoes a BBP-type transition behavior. The paper also compares and quantifies the performance gap between the two approaches.

**Strengths:**

1. The paper compares the performance of tensor-based and matrix-based approaches, providing insights into the advantages of tensor-based methods for structured tensor data.

2. This paper provides a rigorous theoretical analysis of the proposed framework.

3. The paper quantifies the performance gap between these two approaches and derives the precise algorithmic threshold of the matrix-based approach.

**Weaknesses:**

1. The paper does not provide a detailed comparison with other state-of-the-art methods for multi-view clustering.

2. The paper does not include experimental results on real-world datasets to validate its theoretical findings.

3. It is best to provide more detailed experiments to compare the tensor-based method and matrix-based method.

4. It is better to provide a citation for “BBP-type”.

**Questions:**

1. Are the theoretical results of this paper also applicable to tensors with orders greater than 3?

2. How to choose the parameters $(p, n, m)$ in Figure 3?

---

> ### Author Response · Authors · 2023-11-20
>
> Thank you so much for your valuable feedback. We have carefully examined your remarks and hope that our answers will improve the quality of our paper.
>
> **Weaknesses**
> 1. It is important to note that the goal of our work is not to propose a new algorithmic method to solve our estimation problem, but rather to compare from a theoretical standpoint the achievable performances between tensor-based and matrix-based approaches. In this context, experimental comparisons between different algorithms are out of our scope.
> 2. Our main goal is to give rigorous theoretical results quantifying the performance gap between tensor-based and matrix-based approaches for multi-view clustering. However, in all this generality, such a task is quite difficult, and therefore our work considers a simplified model (the particular choice of view function $f_k$) as a first step towards more general rigorous theoretical results. As a result of this simplification, we lack relevant real-world data for the moment. Nevertheless, our theoretical findings give a precise understanding of tensor- and matrix-based approaches, supported by rigorous mathematical proofs, which *per se* do not need validation (the experimental results are only meant to illustrate them).
> 3. Please note that the main goal is to provide a theoretical analysis of the performance gap, with the numerical experiments serving only as illustrations of our findings. In this regard, we believe that the current results already cover the main aspects of our analysis. Could you please point out which additional comparison would be useful?
> 4. A reference for "BBP-type" has been added to the abstract (Baik et al., 2005, "Phase transition of the largest eigenvalue
> for nonnull complex sample covariance matrices").
>
> **Questions**
> 1. In principle, extensions of the model to higher-order cases, such as one in which the signal is given by $\mathbf{M} \otimes \mathbf{z}_1 \otimes \mathbf{z}_2$, can be handled with the same tools and pose no conceptual difficulty. In particular, a matrix-base spectral (i.e., unfolding) method can be similarly employed (see, e.g., Gérard Ben Arous, Daniel Zhengyu Huang, and Jiaoyang Huang, "Long Random Matrices and Tensor Unfolding"). However, we are unaware of any machine learning problem which might motivate such an extension, and for that reason we did not discuss it in the manuscript.
> 2. In practice, these parameters are not chosen, they depend on the data: there are $n$ points of dimension $p$ split into two clusters that are seen from $m$ different views. In the particular setting of Figure 3, we have chosen realistic values of $p$, $n$, and $m$ to perform the simulations.

---

> > ### Comment · Reviewer_9jwW · 2023-11-23
> >
> > Thank you for your detailed response. I appreciate your efforts in addressing my concerns. I will keep my score.

---

### Official Review · Reviewer_xYze · 2023-11-05

**Soundness:** 3 good
**Presentation:** 3 good
**Contribution:** 3 good
**Rating:** 8
**Confidence:** 5

**Summary:**

The paper addresses the *nested matrix tensor model* which is the simplest statistical model for the multi-view clustering problem. The multi-view clustering problem assumes that several *views* of a set of clustered points are available. Each view being a function of the input points. Here we consider that a view is defined as a rescaled version of the input signal, up to additive independent noise.
The paper studies a solution of the simplified "nested matrix tensor" problem, relying on a tensor unfolding schema. Authors provide theoretical results for the proposed solution. The theoretical results borrow tools from random matrix and random tensor theory. REsults allow to identify regimes where the approach works or fails.

**Strengths:**

The papers focus is on an interesting theoretical model for a challenging and relevant application. Authors do a good job motivating the work and stating it both in comparison with theoretical related work and applied related work. The contribution of the paper is insightful and results are non-trivial. Authors have done a good job presenting the main results and providing numerical simulations and graphics that make the findings more accessible.

**Weaknesses:**

Gaps between info-theoretical and computational results are non-trivial in tensor problems and I may have missed it, but I could not see it discussed in this paper (see question).

**Questions:**

When authors mention "impossible" or "possible" recovery, it is unclear to me whether they mean "information theoretically (im)possible" or "computationally (im)possible". Even though in the matrix case these two match, in random tensor problems there are different asymptotics for the two. It seems to me that authors are not dealing with info theoretical phase transitions but with numerical schemas: "assume we do unfolding and apply the algorithm proposed, we find a solution in this regime". Is this correct?

In a regime where SNR is high, authors mention that one can get good results with a simple tensor power iteration (no unfolding). What if we have a good enough guess for the initialization vector of the tensor power iteration? Are these two cases of any practical interest?


While authors mention that coefficients c_i are positive and "This models the fact that, in practice, we deal with large tensors whose dimensions have comparable sizes", I wonder if it could help the paper to also highlight the (easier) cases where n_3 --> infty while others are constant etc. and provide intuitions on what each of these easier cases mean (many views of the problem makes it easy to identify signal). These simple cases may appear trivial from the viewpoint of the developed theory here, but some may be insightful / useful in practice to know about.

From the plot it appears that the main bottleneck is the matrix SNR rho_M. Do we think that in practice (for the multiview clustering problem) it is relevant to consider the tensor formalism to get the bit of signal left in the tensor and get it back through unfolding?

---

> ### Author Response · Authors · 2023-11-20
>
> We would like to sincerely thank you for your valuable feedback. We hope that our answers will fully address your questions.
> 1. The Reviewer is right, thank you for raising this point. Accordingly, we have added a remark in Section 3.1 to avoid any confusion in this sense.
> 2. We assume the Reviewer refers to our comment in the Conclusion. Please note that it already mentions the use of power iteration method initialized by a good enough guess, namely, that given by the unfolding method (that is, the dominant singular vectors of the unfoldings). This combination of methods (unfolding + power iteration) is what is usually done in practice, and can bridge the gap between the dashed lines and the dash-dotted lines of Figure 3, since then the tensor structure is taken into account by the second step.
> 3. In fact, results presented in section 3 remain valid as long as $n_1$ and $n_2$ grow at the same rate. If one wants to consider, as the Reviewer proposes, a case where $n_3 \gg n_1, n_2$, one must only be careful to the fact that $\rho_T = \beta_T^2 n_T / \sqrt{n_1 n_2 n_3}$ must converge to a positive quantity, which changes the non-trivial regime! Here, we would have $\beta_T \ll n_T^{1 / 4}$ in the non-trivial regime, so the case $\beta_T = \Theta(n_T^{1 / 4})$ dealt with in our work corresponds to the limit $\rho_T \to +\infty$ when one considers $n_3 \gg n_1, n_2$. Due to these subtlelties, we are unsure whether such a remark would be productive (without spending significant space explaining this point, it could lead to some confusion). Hence we choose not to add it.
> 4. We are not sure that we understood your question. Can you please reformulate it? The "tensor approach" and the (matrix) "unfolding approach" are two distinct methods, which cannot be combined, to solve the estimation problem considered in our work.

---

### Official Review · Reviewer_yQ9E · 2023-11-09

**Soundness:** 3 good
**Presentation:** 2 fair
**Contribution:** 3 good
**Rating:** 6
**Confidence:** 4

**Summary:**

This paper quantifies the performance gap between the nested matrix-tensor model and its unfolding variant in multi-view clustering.   Specifically, the authors theoretically analyze the alignment level between the leading singular vector learned by the matrix-tensor model and the true signals. Their theoretical results give some interesting insights into tensor-based multi-view clustering. Finally, these results are verified by numerical experiments.

**Strengths:**

1. This paper gives a vital result, i.e., under the nested matrix-tensor model, the analysis of the unfolding method shows the theoretical superiority of the original tensor-based method in terms of accuracy.
2. The proposed theoretical results are interesting and can advance the understanding of tensor data processing in multi-view clustering.

**Weaknesses:**

1. The nested matrix-tensor model can only handle the multi-view clustering task in which the dimensionality of each view is the same. In real-world multi-view datasets, the dimensionality of each view is usually different. I wonder how this issue can be addressed in the nested matrix-tensor model.
2. In the summaries of the main contributions of Page 2, the authors can point out the corresponding theorems at the end of each contribution.
3. There is an error in the definition of $T^{(1)}$. Its element should be $T^{(1)}_{i,n_3 (j-1) +k }$.
4. What is the definition of "SNR"?
5. The significance of all variables in Eq. (2) should be specified for readability.
6. What is the definition of the inner product of two tensors?
7. All theorems lack necessary remarks. It critically hurts the reader's understanding of the proposed results. For example, in Theorem 2, $\zeta$ reflects the alignment level between $\hat{y}$ and $y$. However, $\zeta$ is equal to a complex formula. I can't understand how it can be close to $1$.
8. The detailed deduction of the last equation on Page 17 should be presented. It seems not apparent.
9. Cauchy's integral formula on Page 20 is not in a standard form. The authors should give the relevant references.
10. Some important multi-view clustering literature [1], [2] should be added.
[1] Efficient and Effective Incomplete Multi-view Clustering.TPAMI, 2021.
[2] SimpleMKKM: Simple Multiple Kernel K-means. TPAMI, 2022.

**Questions:**

See the previous box.

**Details Of Ethics Concerns:**

No.

---

> ### Author Response · Authors · 2023-11-20
>
> We are truly grateful to the time and dedication you have given to our work, which has brought us many avenues for improvement. We hope that our answers will plainly address your reviews.
> 1. We concur with the reviewer's observation that, in real-world applications, the dimensionality often varies across different views. In our work, our main goal is to give rigorous theoretical results quantifying the performance gap between tensor-based and matrix-based approaches for multi-view clustering, which is a difficult task in all its generality. Therefore, we study the simpler model with proportional views of the *same size* as a first step towards more general rigorous theoretical results. Further work is needed to rigorously extend our conclusions beyond the considered model.
> 2. References to the corresponding theorems have been added in the summary of contributions.
> 3. This has been corrected, thank you for your careful reading.
> 4. The acronym "SNR" is used only twice in the last paragraph of page 4. In the previous paragraphs it was explicitely written "signal-to-noise ratio" to indicate the strength of the signal relatively to the noise. In the considered model, this is controled by the two parameters $\beta_M$ and $\beta_T$. This is explained in the introduction of Section 3. We have added a clarification on the use of the acronym "SNR".
> 5. We have added a sentence explaining the significance of all variables in equation (2).
> 6. Given to tensors $\mathbf{T}, \mathbf{T}' \in \mathbb{R}^{n_1 \times n_2 \times n_3}$, their tensor product is $\langle \mathbf{T}, \mathbf{T}' \rangle = \sum_{i, j, k = 1}^{n_1, n_2, n_3} T_{i, j, k} T'_{i, j, k}$. This definition has been added to Section 2.2, where tensor notations are introduced.
> 7. We acknowledge that the formula for $\zeta$ given in Theorem 2 is complex. However, Figure 2 precisely shows the behavior of the quantity $\zeta$ depending on the values of $\beta_M$ and $\rho_T$. We can see that $\zeta$ is close to $1$ only if both SNR parameters are large enough (i.e., far from the phase transition given by the red curve $\zeta = 0$). Another way to see it is by taking $\rho_T = \Theta(\beta^2)$ and $\beta_M = \Theta(\beta)$. Then, from the formula in Theorem 2, $\zeta$ behaves like $1 - \frac{1}{\beta^4} \left[ \left( \frac{\beta^2}{\beta^4} \right)^2 + \beta^2 \right]$, which clearly goes to $1$ as $\beta \to +\infty$ (corresponding to the top right corner of Figure 2).
> 8. Thank you for taking the time of reading of the Appendix as well. We have added details on the deduction of this equation: $Q^{-1} Q = I_{n_2}$ and $Q^{-1} = T^{(2)} T^{(2) \top} - s I_{n_2}$, we use the expression of $T^{(2)}$ given in equation (3) and take the expectation to obtain the formula given at the end of page 17.
> 9. The expression of the alignments relies on the eigendecomposition $\tilde{Q}(\tilde{s}) = \sum_{i = 1}^n \frac{1}{\tilde{\lambda} - \tilde{s}} u_i u_i^\top$ where $\tilde{\lambda}_i$ and $u_i$ are eigenvalues and eigenvectors of $\frac{n_T}{\sqrt{n_1 n_2 n_3}} T^{(2)} T^{(2) \top} - \frac{n_2 + n_1 n_3}{\sqrt{n_1 n_2 n_3}} I_n$. Hence, we isolate the spike $\hat{y} \hat{y}^\top$ (corresponding to some $u_i u_i^\top$) using the standard Cauchy's integral formula with a contour circling around the spike eigenvalue only. We have explained this in the paper as well.
> 10. Thank you for the relevant references. They have been added to our "Related work" paragraph.

---

### Meta-Review · Area_Chair_U9RV · 2023-12-03

**Metareview:**

This paper provides theoretical evidence for the empirical observation that tensor-based methods surpass matrix-based methods in multi-view clustering. After the author’s rebuttal, the majority of the reviewers agree to accept the paper, and only one reviewer (who didn’t response to the author’s rebuttal) remains negative. I have also read the paper, and I think the paper has a good shape and could meet the standard of this conference. Hence, I would recommend accepting the paper.

**Justification For Why Not Higher Score:**

The paper suffers from some minor issues, as pointed out by the reviewers.

**Justification For Why Not Lower Score:**

The contribution of the paper is solid and significant.

---

### Decision · Program_Chairs · 2024-01-16

Accept (poster)